# MULTI-MODAL BRAIN ENCODING MODELS FOR MULTI-MODAL STIMULI

**Subba Reddy Oota**[1*]**, Khushbu Pahwa**[2*]**, Mounika Marreddy**[3]**, Maneesh Singh**[4]
**Manish Gupta**[5]**, Bapi S. Raju**[6]
[1]Technische Universität Berlin, Germany, [2]Rice Univ, USA, [3]Univ of Bonn, Germany
[4]Spector Inc, USA, [5]Microsoft, India, [6]IIIT Hyderabad, India
subba.reddy.oota@tu-berlin.de, gmanish@microsoft.com, raju.bapi@iiit.ac.in

## ABSTRACT

Despite participants engaging in *unimodal stimuli*, such as watching images or silent videos, recent work has demonstrated that multi-modal Transformer models can predict visual brain activity impressively well, even with incongruent modality representations. This raises the question of how accurately these multi-modal models can predict brain activity when participants are engaged in *multi-modal stimuli*. As these models grow increasingly popular, their use in studying neural activity provides insights into how our brains respond to such multi-modal naturalistic stimuli, i.e., where it separates and integrates information across modalities through a hierarchy of early sensory regions to higher cognition (language regions). We investigate this question by using multiple unimodal and two types of multi-modal models—cross-modal and jointly pretrained—to determine which type of model is more relevant to fMRI brain activity when participants are engaged in watching movies (videos with audio). We observe that both types of multi-modal models show improved alignment in several language and visual regions. This study also helps in identifying which brain regions process unimodal versus multi-modal information. We further investigate the contribution of each modality to multi-modal alignment by carefully removing unimodal features one by one from multi-modal representations, and find that there is additional information beyond the unimodal embeddings that is processed in the visual and language regions. Based on this investigation, we find that while for cross-modal models, their brain alignment is partially attributed to the video modality; for jointly pretrained models, it is partially attributed to both the video and audio modalities. This serves as a strong motivation for the neuroscience community to investigate the interpretability of these models for deepening our understanding of multi-modal information processing in brain. We make the code publicly available[1].

## 1 INTRODUCTION

Brain encoding aims at predicting the neural brain activity recordings from an input stimulus representation. Recent brain encoding studies use neural models as a powerful approach to better understand the information processing in the brain in response to naturalistic stimuli (Oota et al., 2024b). Current encoding models are trained and tested on brain responses captured from participants who are interacting with *unimodal stimuli*. Several unimodal pretrained models have been used to obtain stimulus representations for this purpose, such as language (Wehbe et al., 2014; Jain & Huth, 2018; Toneva & Wehbe, 2019; Caucheteux & King, 2022; Schrimpf et al., 2021; Toneva et al., 2022; Aw & Toneva, 2023; Oota et al., 2022a), vision (Yamins et al., 2014; Eickenberg et al., 2017; Schrimpf et al., 2018; Wang et al., 2019) or speech (Millet et al., 2022; Vaidya et al., 2022; Tuckute et al., 2023). In this paper, we build encoding models where participants are engaged with *multi-modal stimuli* (e.g., watching movies that include audio). We explore multi-modal stimulus representations extracted using Transformer-based (Vaswani et al., 2017) multi-modal models. Our analysis focuses on brain alignment—the degree of similarity when predicting brain activity using both uni-modal and multi-modal models.

---

[1]https://github.com/subbareddy248/multi-modal-brain-stimuli

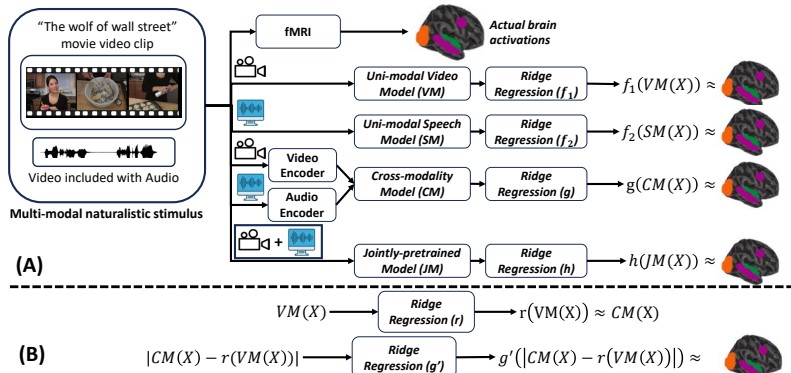

Figure 1: **(A) Overview of our proposed Multi-modal Brain Encoding Pipeline.** Using fMRI recordings from participants watching popular movies included with speech, we align stimulus representations with brain recordings through ridge regression. For uni-modal alignment, we use representations from video models (*VM*) or speech models (*SM*), where the input consists exclusively of either videos (without speech) or speech, respectively. For multi-modal alignment, we leverage representations from cross-modal (*CM*) and jointly-pretrained models (*JM*), where the input consists of both video and speech. Here, $f_1$, $f_2$, $g$ and $h$ are ridge regression models. **(B) Residual Analysis.** First, we remove the uni-modal video model (*VM*) representations from the cross-modal (*CM*) representations by learning a simple linear function $r$ that maps *VM* representations to the *CM* representations, and use this estimated function to obtain the residual representations $|CM(X)-r(VM(X))|$. In step 2, we learn another ridge regression model ($g'$) to measure the brain alignment between residual representations $|CM(X)-r(VM(X))|$ and the fMRI brain recordings. Similarly, residual analysis can also be applied to remove unimodal speech (*SM*) features from *CM* or *JM* representations for a given input *X*.

There is growing evidence that the human brain's ability for multi-modal processing is underpinned by synchronized cortical representations of identical concepts across various sensory modalities (Gauthier et al., 2003; Bracci & Op de Beeck, 2023). Reflecting similar principles, the recent advances in AI systems have led to the development of multi-modal models (like CLIP (Radford et al., 2021), ImageBind (Girdhar et al., 2023), and TVLT (Tang et al., 2022)) use massive interleaved image-text data, speech-text data or video-audio-text data to represent multi-modal input. This recent progress in AI has stimulated advancements in brain encoding models (Conwell et al., 2022); (Doerig et al., 2022; Oota et al., 2022b; Popham et al., 2021; Wang et al., 2023; Tang et al., 2024; Nakagi et al., 2024) that learn effectively from multiple input modalities, despite participants being engaged with unimodal stimulus during experiments, e.g., watching natural scene images, or silent movie clips. However, these studies have experimented with subjects engaged with unimodal stimulus. Only recently there has been some work on training models for true multi-modal stimulus scenarios, but most of them have focused on video+text stimuli (Nakagi et al., 2024; Subramaniam et al., 2024). The only exception being Dong & Toneva (2023a) who experiment with video+audio stimuli and explored brain alignment differences with pretrained versus finetuned joint multi-modal transformer representations.

Using brain recordings of participants watching several popular movies included with audio (St-Laurent et al., 2023), we investigate several research questions. First, we investigate the effectiveness of stimulus representations obtained using multi-modal models versus unimodal models for brain encoding. Multi-modal models are of two broad types: (i) cross-modal pretrained models, where first intermediate representations for both modalities are computed using individual modality encoders and then combined using contrastive loss, and (ii) jointly pretrained models, which involve combining data from multiple modalities at token level itself, and training a single joint encoder. Hence, we also investigate which of the two types (cross-modal versus joint) is better for encoding. We focus on one cross-modal (ImageBind), one jointly pretrained (TVLT), three video and two speech models. Additionally, we explore which modality representations are more brain relevant, and identify which brain regions process unimodal versus multi-modal information. Overall, this research utilizes multi-modal representations to develop encoding models based on fMRI responses (see Fig. 1).

Using our multi-modal brain encoding approach, we examine several insights. First, we use previous neuroscience findings that have identified brain regions involved in visual, language and auditory

processing, and investigate how well our model aligns with these regions when both the model and a human participant watch the same multi-modal video stimuli. Secondly, we hypothesize that multi-modal models capable of learning cross-modal and joint embeddings across various sensory inputs in a manner that mimics brain processing would likely show significant alignment with these neural regions. However, alignment with these brain regions doesn't necessarily mean that the model is effectively learning from multiple modalities, as unimodal models for vision or language or audio have also been shown to significantly align with these brain regions (Wehbe et al., 2014; Toneva et al., 2022; Schrimpf et al., 2021; Millet et al., 2022; Vaidya et al., 2022). To check the second aspect, we investigate this question via a direct approach, closely related to previous studies (Toneva et al., 2022; Oota et al., 2023a;b). For each modality, we analyze how the alignment between brain recordings and multi-modal model representations is affected by the elimination of information related to that particular modality from the model representation.

Our analysis of multi-modal brain alignment leads to several key conclusions: (1) Both cross-modal and jointly pretrained models demonstrate significantly improved brain alignment with language regions (AG, PCC, PTL, and IFG) and visual regions (EVC and MT) when analyzed against unimodal video data. In contrast, compared to unimodal speech-based models, all multi-modal embeddings show significantly better brain alignment, except in the LOC (object visual processing) region. This highlights the ability of multi-modal models to capture additional information—either through knowledge transfer or integration between modalities—which is crucial for multi-modal brain alignment. (2) Using our residual approach, we find that the improved brain alignment in cross-modal models can be partially attributed to the removal of video features alone, rather than auditory features. On the other hand, the improved brain alignment in jointly pretrained models can be partially attributed to the removal of both video and auditory features.

We make the following contributions. (1) To the best of our knowledge, this study is the first to leverage both cross-modal and jointly pretrained multi-modal models to perform brain alignment while subjects are engaged with multi-modal naturalistic stimuli. (2) We evaluate the performance of several unimodal Transformer models (three video and two audio) and measure their brain alignment. (3) Additionally, we remove unimodal features from multi-modal representations to explore the impact on brain alignment before and after their removal. We make the code publicly available[1].

## 2    RELATED WORK

**Multi-modal models.** Pretrained Transformer-based models have been found to be very effective in various tasks related to language (Devlin et al., 2019; Radford et al., 2019), speech (Baevski et al., 2020), and images (Dosovitskiy et al., 2020). To learn associations between pairs of modalities, Transformer models have been pretrained on multiple modalities, showing excellent results in multi-modal tasks like visual question answering and visual common-sense reasoning. These multi-modal models are pretrained in two different ways: (i) cross-modal models that integrate information from multiple modalities and learn a joint encoder, such as VisualBERT (Li et al., 2019) and ImageBind (Girdhar et al., 2023), and (ii) jointly pretrained models like LXMERT (Tan & Bansal, 2019), CLIP (Radford et al., 2021), and TVLT (Tang et al., 2022) which fuse individual modality encoders at different stages, transferring knowledge from one modality to another. In this work, we investigate how the representations extracted from *cross-modal and jointly-pretrained Transformer models* align with human brain recordings when participants engage with multi-modal stimuli.

**Brain encoding using multi-modal models.** The majority of brain encoding studies tend to focus on vision or language alone, in large part due to the availability of only unimodal datasets (Conwell et al., 2022; Wang et al., 2023). Notably, Conwell et al. (2022) conducted controlled comparisons between visual models with identical architecture and training data, finding no performance improvement from contrastive image-language training. Similarly, Wang et al. (2023) reported that language alignment did not enhance encoding performance across most of the high-level visual cortex in their experiments. However, these studies focused on a single modality of input – vision alone. Since human brain perceives the environment using information from multiple modalities (Gauthier et al., 2003), examining the alignment between language and visual representations in the brain by training encoding models on fMRI responses, while extracting joint representations from multi-modal models, can offer insights into the relationship between the two modalities. For instance, it has been shown that multi-modal models like CLIP (Radford et al., 2021) better predict neural responses in the high-level visual cortex as compared to previous vision-only models (Doerig et al., 2022; Wang et al.,

2023; Tang et al., 2024; Nakagi et al., 2024). However, these studies have experimented with subjects engaged with single-modality stimulus, leaving the full potential of these models in true multi-modal scenarios still unclear. Recently, Dong & Toneva (2023b) interpreted the effectiveness of pretraining versus finetuning using a multi-modal video transformer model, *MERLOT Reserve* (Zellers et al., 2022). Further, they leverage single modality embeddings obtained from the same mulitmodal model for residual analysis. It should be noted that multi-modal models can be trained broadly using two strategies – cross-modally (dual) or jointly (Frank et al., 2021). It is conceivable that the representations arising from these two kinds of training might have different brain alignment (Oota et al., 2022b). To address these differences, we experiment with both cross-modal (dual) and joint multi-modal models. In addition, for residual analysis, to avoid any information propagation from one modality to another, we perform a comprehensive study using 3 separate unimodal video and 2 unimodal audio encoders. We discuss more related studies in Appendix J.

## 3 DATASET CURATION

**Brain imaging dataset.** We experiment with a multi-modal naturalistic fMRI dataset, Movie10 (St-Laurent et al., 2023) obtained from the Courtois NeuroMod databank. This dataset was collected while six human subjects passively watched four different movies: *The Bourne supremacy ($\sim$100 mins)*, *The wolf of wall street ($\sim$170 mins)*, *Hidden figures ($\sim$120 mins)* and *Life ($\sim$50 mins)*. Among these, *Hidden figures* and *Life* are repeated twice, with the repeats used for testing and the remaining movies for training. In this work, we use *Life* movie for testing where we average the two repetitions to reduce noise in brain data. This dataset is one of the largest publicly available multi-modal fMRI dataset in terms of number of samples per participant. It includes 4024 TRs (Time Repetitions) for *The Bourne supremacy*, 6898 TRs for *The wolf of wall street* used in train and 2028 TRs for *Life* in test. The fMRI data is collected every 1.49 seconds (= 1 TR).

The dataset is already preprocessed and projected onto the surface space ("fsaverage6"). We use the multi-modal parcellation of the human cerebral cortex based on the Glasser Atlas (which consists of 180 regions of interest in each hemisphere) to report the ROI (region of interest) analysis for the brain maps (Glasser et al., 2016). This includes four visual processing regions (early visual cortex (EVC), object-related areas (LOC), face-related areas (OFA) and scene-related areas (PPA)), one early auditory area (AC), and eight language-relevant regions, encompassing broader language regions: angular gyrus (AG), anterior temporal lobe (ATL), posterior temporal lobe (PTL), inferior frontal gyrus (IFG), inferior frontal gyrus orbital (IFGOrb), middle frontal gyrus (MFG), posterior cingulate cortex (PCC) and dorsal medium prefrontal cortex (dmPFC), based on the Fedorenko lab's language parcels (Milton et al., 2021; Desai et al., 2023). We list detailed sub-ROIs in Appendix C.

**Estimating dataset cross-subject prediction accuracy.** To account for the intrinsic noise in biological measurements, we adapt Schrimpf et al. (2021)'s method to estimate the cross-subject prediction accuracy for a model's performance for the Movie10 fMRI dataset. By subsampling fMRI dataset from 6 participants, we generate all possible combinations of $s$ participants ($s \in [2,6]$) for watching movies, and use a voxel-wise encoding model (see Sec. 5) to predict one participant's response from others. Note that the estimated cross-subject prediction accuracy is based on the assumption of a perfect model, which might differ from real-world scenarios, yet offers valuable insights into model's performance. We estimate cross-subject prediction accuracy by training on the combined brain data from *The Bourne supremacy* and *The wolf of wall street* and testing on the brain data from the movie *Life*. We present the average cross-subject prediction accuracy across voxels for the *Movie10 fMRI* dataset across subjects in Appendix B.

## 4 METHODOLOGY

**Multi-modal Models**: To analyse how human brain process information while engaged in multi-modal stimuli, we use recent popular deep learning models to explore multiple modalities information and build the encoding models in two different ways: "cross-modality pretraining" and "joint pretraining".

**Cross-modality pretrained multi-modal models.** Cross-modality representations involve transferring information or learning from one modality to another. For example, in a cross-modal learning scenario, text descriptions can be used to improve the accuracy of image/video recognition tasks.

This approach is particularly effective when one modality has limited data or indirect relevance to the task, but can be augmented by information from another modality. Recently, a cross-modal model called ImageBind (IB) (Girdhar et al., 2023) has shown immense promise in binding data from six modalities at once, without the need for explicit supervision. ImageBind model uses separate encoders for each individual modality and learns a single shared representation space by leveraging multiple types of image-paired data. ImageBind has 12 layers and outputs a 1024-D representation for each modality.

**Jointly pretrained multi-modal models.** Jointly pretrained multi-modal model representations, on the other hand, involve combining data from multiple modalities to build a more comprehensive joint understanding to improve decision-making processes. The system processes these diverse inputs concurrently to make more informed and robust decisions. TVLT (Tang et al., 2022) is an end-to-end Text-less Vision-Language multi-modal Transformer model for learning joint representations of video and speech from YouTube videos. This joint encoder model consists of a 12-layer encoder (hidden size 768) and uses masked autoencoding objective for both videos and speech. Given the video-speech pairs, the TVLT model gives 768D representations for each modality across 12 layers.

**Extraction of multi-modal features.** To extract video and audio embedding representations from multi-modal models for the brain encoding task, we input video and audio pairs at each TR simultaneously, and obtain the output embeddings for the two modalities from the last layer. Here, we first segment the input video and audio into clips corresponding to 1.49 seconds, which matches the fMRI image rate. For both the models, ImageBind and TVLT, we use the pretrained Transformer weights. ImageBind generates an embedding for each modality (IB video and IB audio) at the output. We refer to IB video, IB-audio as modality-specific embeddings extracted from multi-modal models in the remainder of the paper. We concatenate these embeddings to create what we refer to as IB concat embeddings. On the other hand, TVLT provides a joint embedding across all modalities at each layer. Only for the last layer, TVLT provides an embedding for each modality - referred to as modality specific embeddings extracted from multi-modal models, similar to IB-video and IB-audio.

**Unimodal Models**: To investigate the effectiveness of multi-modal representations in comparison to representations for individual modalities, we use the following methods to obtain embeddings for individual modalities.

**Video-based models.** To extract representations of the video stimulus, we use three popular pretrained Transformer video-based models from Huggingface (Wolf et al., 2020): (1) Vision Transformer Base (ViT-B) (Dosovitskiy et al., 2020), (2) Video Masked Autoencoders (VideoMAE) (Tong et al., 2022) and (3) Video Vision Transformer (ViViT) (Arnab et al., 2021). Details of each model are reported in Table 1 in Appendix.

**Speech-based models.** Similar to video-based models, we use two popular pretrained Transformer speech-based models from Huggingface: (1) Wav2Vec2.0 (Baevski et al., 2020) and (2) AST (Baade et al., 2022). Details of each model are reported in Table 1 in Appendix D.

**Extraction of video features.** ViT-B (Dosovitskiy et al., 2020), the underlying video encoder model for ImageBind is used for extracting representations for all frames in each TR for every video. To extract embedding at each TR, we average all frame embeddings and obtain the corresponding video representation. For VideoMAE and ViViT, we directly obtain the video embeddings for each TR. All 3 models provide 768 dimensional representations and all of them are 12-layer Transformer encoders.

**Extraction of speech features.** To explore whether speech models incorporate linguistic information, we extract representations beyond 1.49 secs, i.e., we considered context window of 16 secs with stride of 100 msecs and considered the last token as the representative for each context window. The pretrained speech-based models output token representations at different layers. Both Wav2Vec2.0 and AST models provide 768 dimensional representations and all of them are 12-layer Transformer encoders. Finally, we align these representations with the fMRI data acquisition rate by downsampling the stimulus features with a 3-lobed Lanczos filter, thus producing chunk-embeddings for each TR.

## 5   EXPERIMENTAL SETUP

**Encoding model.** We train bootstrap ridge regression based voxel-wise encoding models (Deniz et al., 2019) to predict the fMRI brain activity associated with the stimulus representations obtained from the individual modalities (speech and video) and multi-modal embeddings from cross-modal and jointly pretrained multi-modal models. We employ z-score thresholding separately for both input

stimulus representations and brain recordings for training and test datasets. This helps identify and remove extreme outliers that could disproportionately affect the Pearson Correlation results. For each subject, we account for the delay in the hemodynamic response by modeling hemodynamic response function using a finite response filter (FIR) per voxel with 5 temporal delays (TRs) corresponding to $\sim$7.5 seconds (Huth et al., 2022). Formally, at each time step $t$, we encode the stimuli as $X_t \in \mathbb{R}^D$ and brain region voxels $Y_t \in \mathbb{R}^V$, where $D$ denotes the dimension of the concatenation of delayed 5 TRs, and $V$ denotes the number of voxels. Overall, with $N$ such TRs, we obtain $N$ training examples.

**Train-test setup.** We build encoding models where the train and test sets are totally disjoint and the model cannot use any clock relationships from the training data during inference. To be completely clear: independent encoding models are trained for each subject using data concatenated from two movies (*The Bourne supremacy*: 4024 TRs and The *wolf of wall street*: 6898 TRs). The test set consisted only data from the *"Life" movie* (2028 TRs). Thus there is no possibility of any information leakage during inference on the test set. Model details and hyper-parameter settings are in Appendix E.

**Removal of a single modality features from multi-modal representations.** For removing unimodal model representations (*VM* or *SM*) from the multi-modal model representations (*CM* or *JM*), we employ the direct or residual approach, as outlined by Toneva et al. (2022); Oota et al. (2023a); Oota & Toneva (2023); Dong & Toneva (2023b); Oota et al. (2024a). This method estimates the impact of specific modality features on the alignment between the model and brain recordings by comparing the alignment before and after computationally removing the targeted modality features from the multi-modal representations. To remove features corresponding to a particular modality (*VM* or *SM*) from multi-modal model representations, we remove the linear contribution of the unimodal features by training a ridge regression model ($r$), where the unimodal feature vector is the input and the multi-modal representation serves as the target. Since our encoding model (ridge regression) is also a linear function, this linear removal limits the contribution of features for the particular modality to the eventual brain alignment. The approach is illustrated in Fig. 1 (B).

**Evaluation metrics.** We evaluate our models using Pearson Correlation (PC) which is a standard metric for evaluating brain alignment (Jain & Huth, 2018; Schrimpf et al., 2021; Goldstein et al., 2022). Let TR be the number of time repetitions in the test set. Let $Y = \{Y_i\}_{i=1}^{TR}$ and $\hat{Y} = \{\hat{Y}_i\}_{i=1}^{TR}$ denote the actual and predicted value vectors for a single voxel. Thus, $Y \ and \ \hat{Y} \in \mathbb{R}^{TR}$. We use Pearson Correlation (PC) which is computed as $corr(Y, \hat{Y})$ where corr is the correlation function. The final measure of a model's performance is obtained by calculating Pearson's correlation between the model's predictions and neural recordings. To quantify the model predictions, the resulting model prediction correlations are divided by the estimated cross-subject prediction accuracy and averaged across voxels, regions, and participants, resulting in a standardized measure of performance referred to as normalized brain alignment. For calculating normalized alignment, we select the voxels whose cross-subject prediction accuracy is $\geq 0.05$.

**Statistical significance.** To determine if normalized predictivity scores are significantly higher than chance, we run a permutation test using blocks of 10 contiguous fMRI TRs (considering the slowness of hemodynamic response) rather than individual TRs. By permuting predictions 5000 times, we create an empirical distribution for chance performance, from which we estimate p-value of the actual performance. The choice of these specific permutation test configurations is based on established methodologies in previous research (Deniz et al., 2019; Reddy & Wehbe, 2021; Oota et al., 2024a). To estimate the statistical significance of performance differences, such as between the model's predictions and chance or residual predictions and chance, we utilized the Wilcoxon signed-rank test (Conover, 1999), applying it to the mean normalized predictivity for the participants. Finally, the Benjamini-Hochberg False Discovery Rate (FDR) correction for multiple comparisons (Benjamini & Hochberg, 1995) is used for all the tests (appropriate because fMRI data is considered to have positive dependence (Genovese, 2000)). In all cases, we denote significant differences (p$\leq$ 0.05) with a $*$ or $\wedge$.

# 6 RESULTS

## 6.1 HOW EFFECTIVE ARE MULTI-MODAL REPRESENTATIONS?

In Fig. 2, we present the average normalized brain alignment scores for both multi-modal and individual modality features. Specifically, we show the normalized brain alignment for cross-modality

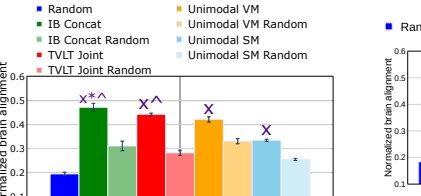 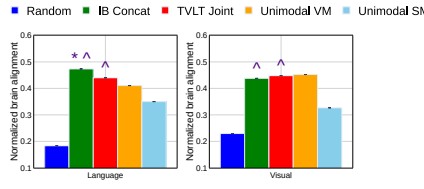

Figure 2: (*Left*) Avg normalized brain alignment of pretrained vs randomly initialized multi-modal and unimodal models across whole brain. × ⟹ pretrained model embeddings are significantly better than randomly initialized models, i.e., p≤ 0.05. (*Right*) Avg normalized brain alignment for both multi-modal and unimodal model features specifically within language and visual regions. Blue bar represents the normalized alignment using randomly generated vector embeddings. Error bars indicate the standard error of the mean across participants. ∗ ⟹ multi-modal embeddings are significantly better than unimodal video models (VM), i.e., p≤ 0.05. ∧ ⟹ multi-modal embeddings are significantly better than unimodal speech models (SM), i.e., p≤ 0.05.

(ImageBind), jointly pretrained multi-modal (TVLT), and the average from individual video and speech models. The results are shown for whole brain (Fig. 2 *Left*), and also for average across language and visual ROIs (Fig. 2 *Right*). Results for individual ROIs are in Fig. 3.

**Baseline comparison.** To compare the brain predictivity of multi-modal and unimodal models against baseline performance, we employ two baselines: (i) randomly generated vector embeddings to predict brain activity, and (ii) randomly initialized models for ImageBind, TVLT, Unimodal VM, and Unimodal SM. Fig. 2 (*Left*) displays the comparison of average normalized brain alignment of randomly generated vectors, pretrained and randomly initialized models. From Fig. 2 (*Left*), we observe that randomly initialized models show significantly better alignment than random vectors. However, the pretrained model embedding brain alignment is significantly better than randomly initialized models. This shows that the representations from these multi-modal models are significant enough for learning non-trivial alignment with the fMRI recordings of multi-modal stimuli.

CROSS-MODAL VS. JOINTLY PRETRAINED MULTI-MODAL MODELS VS. UNIMODAL MODELS.

**Whole brain analysis.** Fig. 2 (*Left*) displays results for whole brain analysis, where the IB Concat bar plot corresponds to results for representations from a cross-modal model, while TVLT Joint bar plot corresponds to results for representations from a jointly pretrained multi-modal model. From Fig. 2 (*Left*), we make the following observations: (i) At the whole brain level, the Wilcoxon signed-rank test shows that the differences in embeddings from the IB Concat and TVLT models are not statistically significant. (ii) The multi-modal embeddings show improved brain alignment compared to unimodal models. Specifically, cross-modal embeddings are significantly better than both unimodal video and speech models, while jointly pretrained embeddings are significantly better than speech models. This implies that cross-modal embeddings contain additional information beyond the two modalities, while embeddings from a jointly pretrained model do not provide extra information beyond unimodal visual information but do contain additional information beyond unimodal speech.

**Whole language and visual region analysis.** We also present average normalized brain alignment results across language and visual regions in Figs. 2 (*Right*). The Wilcoxon signed-rank test shows that the differences in embeddings from the IB Concat and TVLT models are not statistically significant when averaged across language and visual regions. Similar to whole brain performance, in the language regions, cross-modal embeddings are significantly better than both unimodal video and speech models, while jointly pretrained embeddings are significantly better than unimodal speech models. In contrast, for the visual regions, the normalized brain alignment of cross-modal and jointly pretrained embeddings is similar to the performance of unimodal video models. This implies that when we average across visual regions, there is no additional information beyond unimodal video features. However, when compared to unimodal speech features, both multi-modal embeddings show significant improvement.

**ROI-Level analysis of joint embeddings from multi-modal models.** Since we didn't observe any significant difference at the whole brain level and when averaged across language and visual regions, between cross-modal and jointly pretrained multi-modal models, we attempt to seek if there are any differences when we pay a closer look at the individual ROIs. We present results for language regions such as Angular gyrus (AG), the posterior temporal lobe (PTL), and the inferior frontal gyrus (IFG) in Fig. 3. Additionally, we cover visual regions like early visual cortex (EVC), scene visual areas (PPA)

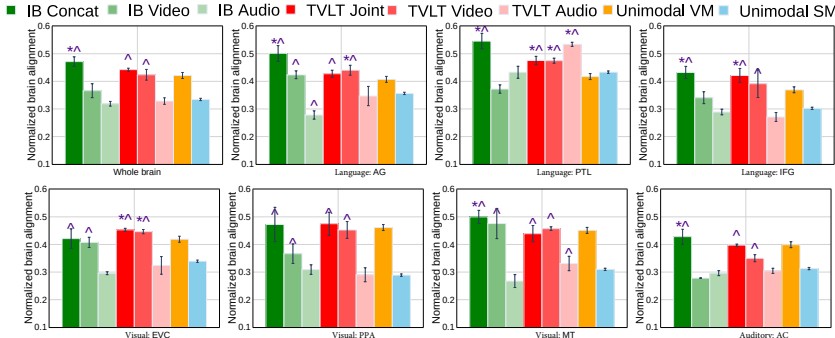

Figure 3: Average normalized brain alignment for video and audio modalities from multi-modal and individual modality features across whole brain and several ROIs of language (AG, PTL and IFG), visual (EVC, PPA and MT) and auditory cortex (AC). Error bars indicate the standard error of the mean across participants. ∗ indicates cases where multi-modal embeddings are significantly better than unimodal video models (VM), i.e., p≤ 0.05. ∧ indicates cases where multi-modal embeddings are significantly better than unimodal speech models (SM), i.e., p≤ 0.05

and middle temporal gyrus (MT), as well as early auditory cortex (AC). In this figure, we also report the average normalized brain alignment of each modality obtained from multi-modal models. Unlike the whole brain analysis, we observe some differences between cross-modal and jointly pretrained models in several language and visual ROIs. Results for other ROIs are in Fig. 7 in Appendix F. Our observations are as follows: (i) Cross-modal IB Concat embeddings are significantly better than TVLT Joint embeddings in semantic regions such as AG and PCC, as well as the multi-modal processing region MT. (ii) Conversely, TVLT Joint embeddings are significantly better than IB Concat embeddings in dmPFC regions. This indicates that while both cross-modal and jointly pretrained multi-modal models perform similarly at a macro level, there are individual differences at micro level. This observation motivated us to do further detailed analysis in Sec. 6.2 and 6.3. Results for each unimodal video and each unimodal speech model for individual ROIs are in Fig. 8 in Appendix F.

**ROI-Level analysis of modality-specific embeddings extracted from multi-modal models.** While considering both joint and each modality embeddings from multi-modal models, we make the following observations from Fig. 3: (1) Cross-modal IB video embeddings exhibit improved brain alignment compared to unimodal video in the AG and MT regions with the exceptions of the PTL and AC regions. Notably, in the AC, which is an early auditory processing area primarily handling sound-related information rather than higher cognitive functions (such as language processing), audio embeddings yield higher brain predictivity than video embeddings. Interestingly, this differential effect is not observed when comparing ImageBind audio embeddings to unimodal audio embeddings. These findings suggest that video modality information contributes more significantly to brain alignment in the context of ImageBind concatenated embeddings derived from cross-modal models. (2) TVLT video embeddings show improved brain alignment in the AG, PTL, PCC, dmPFC and EVC regions, with other regions displaying similar normalized brain alignment unimodal video embeddings. Interestingly, while the PTL is known for auditory processing, it also contributes to the integration of visual and auditory inputs. The improved alignment of video embeddings here indicates that visual information enhances the processing capabilities in this region. (3) Consistent with the cross-modality models, in jointly pretrained TVLT models, TVLT video embeddings significantly outperform TVLT audio embeddings, except in PTL region. These observations indicate that video information is advantageous for both cross-modal and jointly pretrained models, whereas audio embeddings mainly benefit the PTL region.

## 6.2 WHICH BRAIN REGIONS PROCESS UNI- AND MULTI-MODAL INFORMATION?

From Fig. 3, we observe that multi-modal video embeddings exhibit improved brain alignment not only in the whole brain but also in various language, visual and multi-modal regions. For instance, the cross-modal IB Concat embeddings demonstrate superior brain alignment compared to unimodal video-based models in areas such as the AG, PTL, IFG, and PCC. Moreover, TVLT-joint embeddings show notable enhancements in the AG, PTL, IFG, PCC, dmPFC and EVC regions. In contrast, compared to unimodal speech-based models, all multi-modal embeddings display significantly better brain alignment, except the LOC (object visual processing) region. The LOC region is highly

Figure 4: Residual analysis: Average normalized brain alignment was computed across participants before and after removal of video and audio embeddings from both jointly pretrained and cross-modality models. Error bars indicate the standard error of the mean across participants. "-" symbol represents residuals.

specialized for processing visual information related to object recognition. When audio information is integrated, the specific visual features crucial for LOC region alignment may become less pronounced in the embeddings, leading to slightly reduced alignment compared to unimodal visual models. However, overall, our findings suggests that integrating multiple modalities leads to transferring information from one modality to another, resulting in improved brain predictability. Hence, it can be inferred that multi-modal models can indeed learn multi-modal linkages that are relevant to the brain.

When subjects engage with multi-modality stimuli, we observe that multi-modal embeddings show improvements in semantic regions such as the AG, PCC and dmPFC, and syntactic regions such as the PTL and IFG. Overall, we find that multi-modal information is processed in only a few regions. Furthermore, several regions, including the PPA (scene visual area), EVC (early visual cortex), ATL (anterior temporal lobe), IFGOrb, MFG, and dmPFC, exhibit similar brain alignment with both unimodal and multi-modal embeddings.

### 6.3 HOW DOES EACH MODALITY CONTRIBUTE TO THE MULTI-MODAL BRAIN ALIGNMENT?

To understand the contribution of each modality to the multi-modal brain alignment for multi-modality naturalistic stimulus, we perform residual analyses by removing the unimodal model features from multi-modal joint representations as well as multi-modal video or audio representations from joint representations and measure the differences in brain alignment before and after removal modality-specific features. To check the quality of information removal using residual analysis, We computed Pearson correlation where unimodal video features are projected onto the multi-modal IB Concat feature space using the residual approach. We found correlation to be as low as 0.56 which implies that unimodal video features are successfully removed from multi-modal representations. Fig. 4 shows normalized alignment for language (AG) and visual regions (MT).

**Cross-modal multi-modal models.** The alignment in regions AG and MT is extremely high, and this alignment is only partially explained by video features. This implies that significant unexplained alignment remains after the removal of video features. Conversely, the removal of speech features does not lead to a drop in brain alignment, indicating that there is additional information beyond speech features that is processed in these regions. This means that in cross-modal models, when transferring knowledge from one modality to another, the model relies more heavily on visual information. As a result, the model becomes more focused on video inputs rather than audio inputs. This likely reflects the model's preference for using the detailed visual features that align closely with brain activity in regions AG and MT, leading to the observed high alignment.

**Jointly pretrained multi-modal models.** The alignment in regions AG and MT is extremely high, and this alignment is partially explained by both video and audio features. Unlike cross-modal representations, the TVLT model learns a more balanced representation of both video and audio features. This leads to integrated information from both modalities, making the model less sensitive to the loss of features from a specific modality. As a result, we observe only a small drop in brain alignment when either modality is removed. This suggests that the model is capturing more high-level abstract and semantic information that goes beyond the specific features of just one modality. We observe similar findings for language ROIs such as PTL, MFG, ATL, PCC and visual regions EVC, LOC and OFA, as shown in Figs. 9 and 10 in Appendix G. These results suggest that there is additional information beyond the unimodal embeddings considered in this study that is processed in the visual and language regions.

**Qualitative analysis.** We compute the percentage decrease in alignment for each voxel following the removal of unimodal video embeddings from the IB Concat (cross-modality) and the TVLT Joint

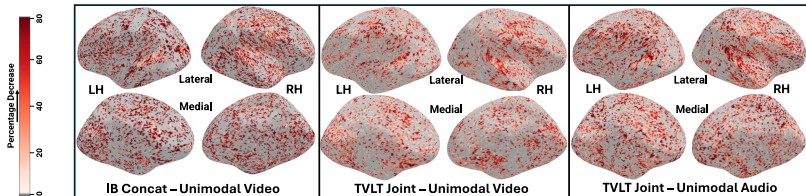

Figure 5: Percent decrease of brain alignment after removal of unimodal embeddings from different multimodal models. (Left) Removal of unimodal VM embeddings from IB-Concat. (Middle) Removal of unimodal VM embeddings from jointly pretrained TVLT. (Right) Removal of unimodal SM embeddings from TVLT Joint. The color bar indicates the percent of decrease where darker shade of red denotes higher and white denotes zero. LH: Left Hemisphere and RH: Right Hemisphere.

(jointly pretrained model), with projections onto the brain surface averaged across participants, as depicted in Fig. 5. The colorbar shows the percentage decrease in brain alignment, where red voxels indicate a higher percentage decrease and white voxels indicate areas where unimodal video features do not contribute any shared information within the multi-modal context. We observe that removal of unimodal video features leads to a significant drop (40-50%) in performance in the visual regions for IB Concat, and in language regions (PTL & MFG) for TVLT Joint.

# 7    DISCUSSION

Using multi-modal model representations, including both cross-modal and jointly pretrained types, we evaluated how these representations can predict fMRI brain activity when participants are engaged in multi-modality naturalistic stimuli. Further, we compared both multi-modal and unimodal representations and observed their alignment with both unimodal and multi-modal brain regions. This is achieved by removing information related to unimodal stimulus features (audio and video) and observing how this perturbation affects the alignment with fMRI brain recordings acquired while participants are engaged in watching multi-modality naturalistic movies.

Our analysis of multi-modal brain alignment yields several important conclusions: (1) The improved brain alignment of the multi-modal models over unimodal models, across several language, visual, and auditory regions is only partially attributable to the video and audio stimulus features presented to the model. A deeper understanding of these models is required to shed light on the underlying information processing of both unimodal and multi-modal information. (2) Cross-modal representations have significantly improved brain alignment in language regions such as AG, PCC and PTL. This variance can be partially attributed to the video features alone, rather than removal of auditory features. (3) Video embeddings from multi-modal models exhibit higher brain alignment than audio embeddings, except in the PTL and AC regions. This suggests that audio-based models may encode weaker brain-relevant semantics, as similar findings are observed in a recent study (Oota et al., 2024a). (4) Both cross-modal and jointly pretrained models demonstrate significantly improved brain alignment with language regions (AG, PCC, PTL and IFG) compared to visual regions when analyzed against unimodal video data. In contrast, when compared to unimodal audio-based models, all multi-modal embeddings display significantly better brain alignment, with the exception of the LOC region. This underscores the capability of multi-modal models to capture additional information—either through knowledge transfer or integration between modalities—crucial for multi-modal brain alignment.

The model training protocol of TVLT appears more in line with how humans learn during development when they experience multiple modalities simultaneously and the learning is mediated by the experience of joint inter-modal associations. It is unlikely that the human system experiences these modalities in isolation, except in cases of congenital conditions where the inputs from a specific modality are not accessible. Given that the brain alignment observed in TVLT model in a language region like AG is less sensitive to loss of information from specific modalities, we believe that AG serves as a multi-modal convergent buffer integrating spatio-temporal information from multiple sensory modalities to process narratives (Dong & Toneva, 2023b; Humphreys & Tibon, 2023). The results of high alignment found in AG even in IB-Concat but more brittle with respect to loss of information from a specific modality are also interesting. It would be interesting to study patterns of activation in AG in patients who acquired visual or auditory function later in their life (Hölig et al., 2023) to see if one observes such brittleness in the representations acquired. We make the code publicly available[1]. Lastly, we discuss limitations of our work in Appendix O.

## 8 ETHICS STATEMENT

We did not create any new neural recordings data as part of this work. We used the Movie10 dataset which is publicly available without any restrictions. Movie10 dataset can be downloaded from `https://github.com/courtois-neuromod/movie10/tree/33a97c01503315e5e09b3ac07c6ccadb8b887dcf`. Please read their terms of use[2] for more details.

We do not foresee any harmful uses of this technology.

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

## A    OVERVIEW OF APPENDIX SECTIONS

- Section B: Cross-subject prediction accuracy
- Section C: Detailed sub-ROIs of language, visual and auditory regions
- Section D: Details of pretrained Transformer models
- Section E: Implementation details for reproducibility.
- Section F: Effectiveness of multi-modal vs unimodal representations for various brain regions
- Section G: How is the brain alignment of multi-modal features affected by the elimination of a particular modality?
- Section H: Layerwise brain alignment
- Section I: Why the choice of ridge regression instead of more complex machine learning models?
- Section J: Extended Related Works
- Section K: Baseline Analysis: Scrambling Inputs to Multi-modal Models
- Section L: Whole Brain, Language and Visual ROIs analysis: Shared and Unique variance between Multi-modal and Unimodal models
- Section M: Multi-modal versus unimodal effects
- Section N: Impact of diverse model architectures on performance comparison
- Section O: Limitations.

## B    CROSS-SUBJECT PREDICTION ACCURACY

We estimate cross-subject prediction accuracy in three settings: (i) training with *The Bourne supremacy* and testing with *Life* data, (ii) training with *The wolf of wall street* and testing with *Life* data, and (iii) training with both *The Bourne supremacy* and *The wolf of wall street* and testing with Life data. We present the average cross-subject prediction accuracy across voxels for the *Movie10 fMRI* dataset and across the three settings in Fig. 6.

## C    DETAILED SUB-ROIS OF LANGUAGE, VISUAL AND AUDITORY REGIONS

The data covers seven brain regions of interest (ROIs) in the human brain with the following subdivisions: (i) early visual (EV: V1, V2, V3, V3B, and V4); (ii) object-related areas (LO1 and LO2); (iii) face-related areas (OFA), (iv) scene-related areas (PPA), (v) middle temporal (MT: MT, MST, LO3, FST and V3CD), (vi) late language regions, encompassing broader language regions: angular gyrus (AG: PFm, PGs, PGi, TPOJ2, TPOJ3), lateral temporal cortex (LTC: STSda, STSva, STGa, TE1a, TE2a, TGv, TGd, A5, STSdp, STSvp, PSL, STV, TPOJ1), inferior frontal gyrus (IFG: 44, 45, IFJa, IFSp) and middle frontal gyrus (MFG: 55b) (Baker et al., 2018; Milton et al., 2021; Desai et al., 2023).

## D    DETAILS OF PRETRAINED TRANSFORMER MODELS

Details of each pretrained Transformer model are reported in Table 1 in Appendix. From the table, we can clearly observe that both multi-modal models (ImageBind and TVLT) maintain similar backbone architectures for videos but differ in their backbone architecture for embedding audio as well as in the training strategies. For the TVLT model, video embeddings are captured from the ViT model, while audio embeddings are generated from Mel Spectrograms and jointly pretrained within a single Transformer encoder. In contrast, the ImageBind model uses the ViT model as the backbone for Images and Videos, while the AST model is used for Audio; these individual encoders are used and learn a common embedding space.

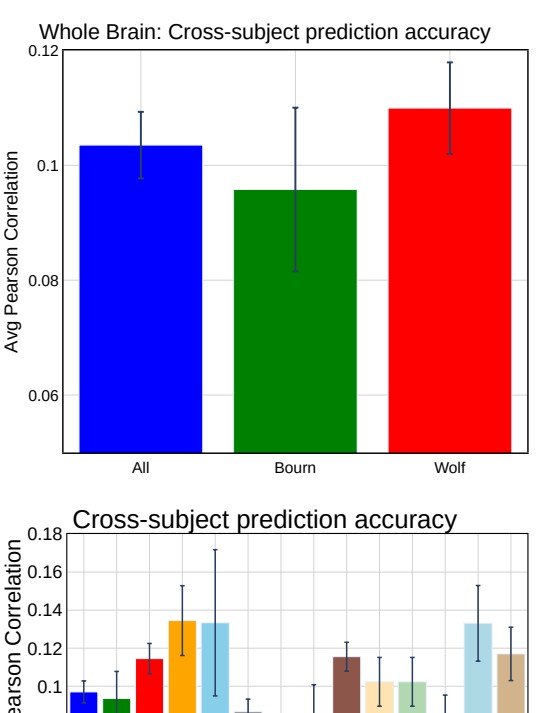

Figure 6: Cross-subject prediction accuracy: (top) across whole brain, (bottom) across language, visual and auditory regions.

Table 1: Pretrained Transformer-based Encoder Models. All models have 12 layers.

| Model Name | Pretraining data modality | Pretraining Method | # Parameters | Dataset | Layers | Backbone |
|---|---|---|---|---|---|---|
| ImageBind | Video & Audio | Cross-model multi-modal Transformer | 132 M | Audioset, Ego4D, SUN RGB-D | 12 | ViT for Images and Videos, AST for audio |
| TVLT | Video & Audio | Jointly pretrained on video and audio (Masked auto encoder) | 88 M | HowTo100M, YTTemporal180M | 12 | ViT for video embeddings, and Spectrogram for audio embeddings |
| ViT-B | Image | Vision Transformer | 86 M | ImageNet | 12 | Transformer encoder |
| VideoMAE | Video | Masked autoencoder for video inputs | 87 M | Kinetics, Epic Kitchens 100, Something-Something v2 | 12 | ViT-B |
| ViViT | Video | Video vision Transformer | 86 M | Kinetics, Epic Kitchens 100, Something-Something v2 | 12 | ViT-B |
| Wav2Vec 2.0-base | Speech | Speech-based Transformer model | 95 M | Librispeech | 12 | Transformer encoder |
| AST | Speech | Audio Spectrogram Transformer | 86 M | AudioSet, ESC-50 and Speech commands | 12 | Initialized with ViT-B weights |

# E    IMPLEMENTATION DETAILS FOR REPRODUCIBILITY.

All experiments were conducted on a machine with 1 NVIDIA GeForce-GTX GPU with 16GB GPU RAM. We used bootstrap ridge-regression (Appendix I) with MSE loss function; L2-decay ($\lambda$) varied from $10^1$ to $10^3$. Best $\lambda$ was chosen by tuning on validation data that comprised a randomly chosen 10% subset from train set used only for hyper-parameter tuning.

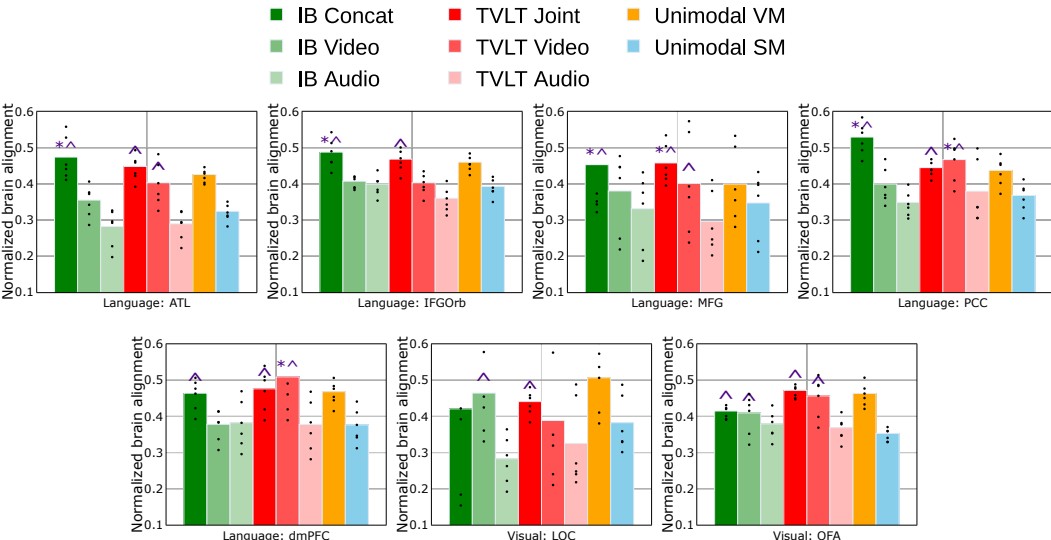

Figure 7: Average normalized brain alignment for per video and audio modalities from multi-modal and individual modality features across whole brain and several ROIs of language (ATL, IFGOrb, MFG, PCC, dmPFC) and visual (LOC, OFA). The points overlaid on the bars represent the normalized brain alignment scores of the six participants.

## F  Effectiveness of multi-modal vs unimodal representations for various brain regions

Average normalized brain alignment for per video and audio modalities from multi-modal and individual modality features across whole brain and several ROIs of language (ATL, IFGOrb, MFG, PCC, dmPFC) and visual (LOC, OFA) are shown in Fig. 7. Our observations are as follows: (i) Cross-modal IB Concat embeddings are significantly better than TVLT Joint embeddings in semantic regions such as AG and PCC, as well as the multi-modal processing region MT. (ii) Conversely, TVLT Joint embeddings are significantly better than IB Concat embeddings in dmPFC regions. This indicates that while both cross-modal and jointly pretrained multi-modal models perform similarly at a macro level, there are individual differences at micro level.

We now present the results for per unimodal video model and per speech model in Fig. 8. Similar to the average results of unimodal video and speech models, we observe that multi-modal models exhibit better normalized brain alignment than individual unimodal video and speech models across language and visual regions. Among unimodal speech models, the AST model shows better normalized brain alignment than the Wav2vec2.0 model. Among unimodal video models, each unimodal video model displays notably consistent performance across regions.

## G  How is the brain alignment of multi-modal features affected by the elimination of a particular modality?

To understand the contribution of each modality to the multi-modal brain alignment for multi-modal naturalistic stimulus, we perform residual analyses by removing the unimodality features from multi-modal joint representations as well as multi-modal video or audio representations from joint representations and measure the differences in brain alignment before and after removal modality-specific features. Figs. 9 and 10 display the normalized brain alignment for language ROIs such as PTL, MFG, ATL, PCC and visual regions EVC, LOC and OFA. We note a decrease in brain alignment for these regions following the removal of video embeddings from cross-modality models, whereas the removal of audio embeddings does not affect the brain alignment. On the other hand, for jointly pretrained models, removal of both video and audio embeddings partially impacts the brain alignment.

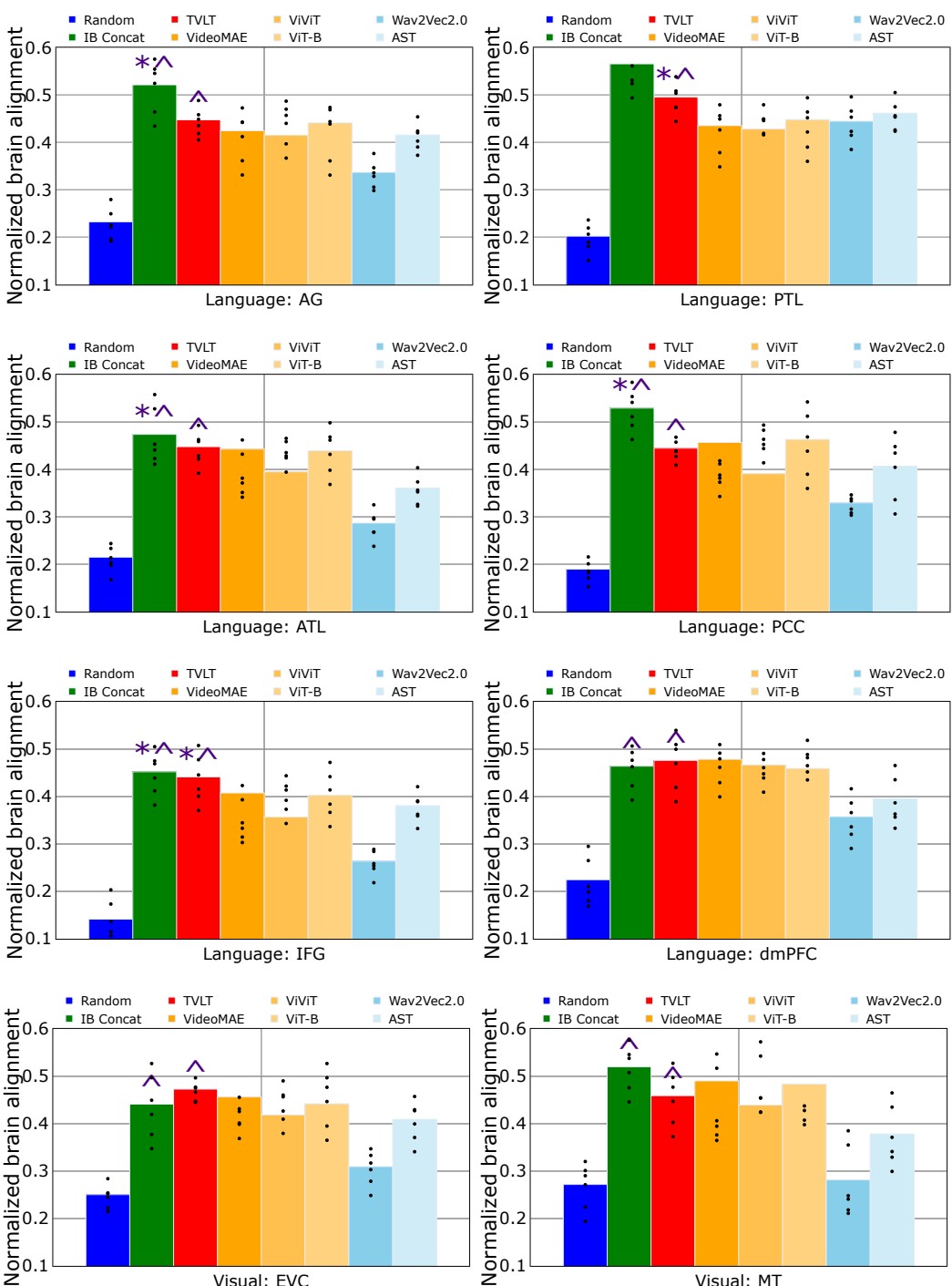

Figure 8: Average normalized brain alignment for video and audio modalities from multi-modal and individual modality features across whole brain and several ROIs of language (AG, ATL, PTL, IFG, PCC, dmPFC) and visual (EVC, MT). Error bars indicate the standard error of the mean across participants.

## H  LAYERWISE BRAIN ALIGNMENT

We now plot the layer-wise normalized brain alignment for the Unimodal models and TVLT joint model, as shown in Fig. 11. Observation from Fig. 11 indicates a consistent drop in performance from early to lower layers, specifically for both TVLT joint and unimodal video models. The key finding

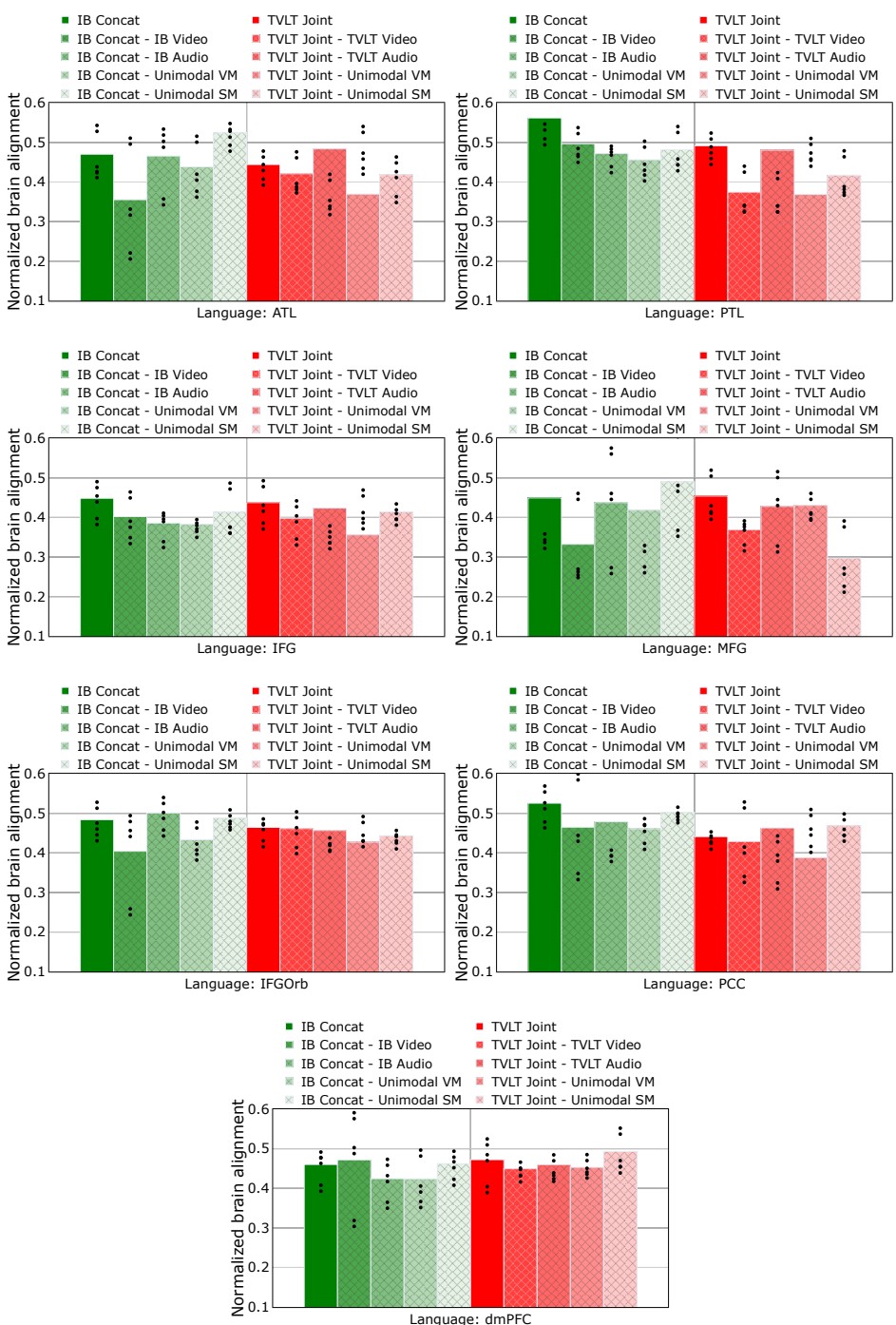

Figure 9: Residual analysis for ATL, PTL, IFG, MFG, IFGOrb, PCC and dmPFC regions: Average normalized brain alignment was computed across participants before and after removal of video and audio embeddings from both jointly pretrained and cross-modality models. The points overlaid on the bars represent the normalized brain alignment scores of the six participants. "-" symbol represents residuals.

here is that our results that TVLT joint embeddings showcase improved brain alignment across all the layers compared to unimodal video and speech embeddings.

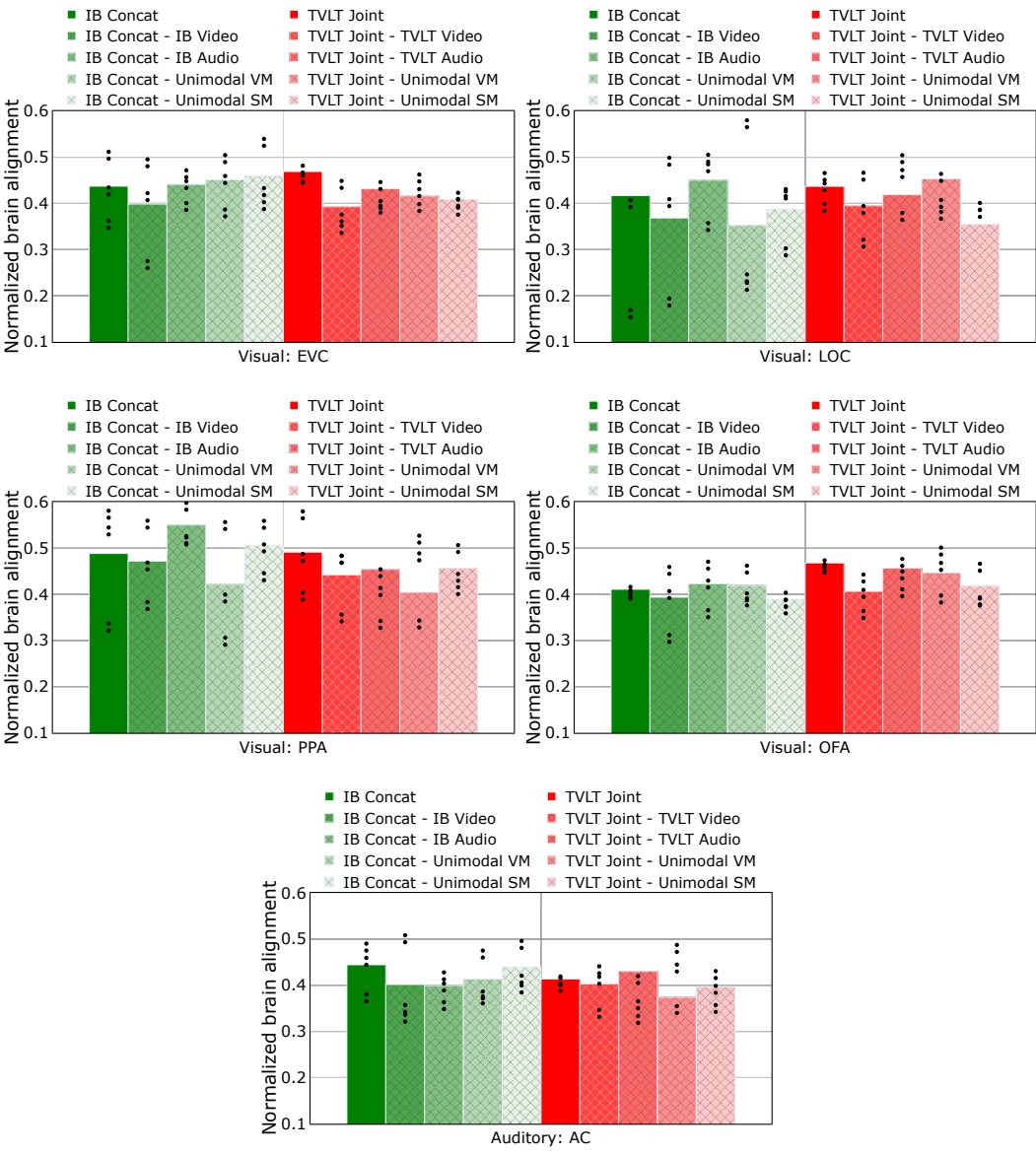

Figure 10: Residual analysis for EVC, LOC, PPA, OFA and AC regions: Average normalized brain alignment was computed across participants before and after removal of video and audio embeddings from both jointly pretrained and cross-modality models. The points overlaid on the bars represent the normalized brain alignment scores of the six participants. "-" symbol represents residuals.

# I    WHY THE CHOICE OF RIDGE REGRESSION INSTEAD OF MORE COMPLEX MACHINE LEARNING MODELS?

Since fMRI brain recordings have a low signal-to-noise ratio, and pretrained language models are trained in a non-linear fashion, the model representations are rich and complex. To understand the relationship between brain activity and various stimuli, a large body of brain encoding literature over the past two decades Wehbe et al. (2014); Huth et al. (2016); Jain & Huth (2018); Toneva & Wehbe (2019); Schrimpf et al. (2021); Caucheteux & King (2022); Antonello et al. (2024); Vaidya et al. (2022); Millet et al. (2022); Oota et al. (2023a; 2024a); Wang et al. (2023) has preferred ridge regression due to its simplicity. Ridge regression is a linear model, making it easier to interpret and understand compared to more complex models. Further, the regularization in ridge regression helps manage the noise effectively, leading to more robust and reliable models.

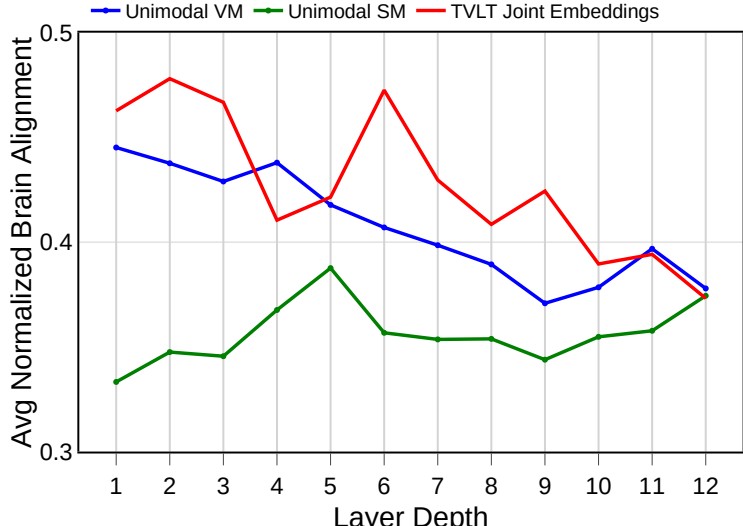

Figure 11: Normalized brain alignment across layers for multi-modal model (TVLT joint embeddings) and unimodal video and speech models.

## J EXTENDED RELATED WORKS

Tang et al. (2024) demonstrate the use of multi-modal models in a cross-modal experiment to assess how well the language encoding models can predict movie-fMRI responses and how well the vision encoding models can predict narrative story-fMRI. Nakagi et al. (2024) analyzed fMRI related to video content viewing and found distinct brain regions associated with different semantic levels, highlighting the significance of modeling various levels of semantic content simultaneously.

Subramaniam et al. (2024) utilized vision-language models based on image frame-text pairs, despite the stimuli being continuous movies. While effective for certain tasks, this approach may overlook the temporal dynamics inherent in videos. Additionally, their use of stereoencephalography (sEEG), although offers high temporal resolution, is limited in spatial resolution and typically restricted to specific brain regions where the electrodes are implanted. Finally, their study focused on multi-modal integration in a cross-modal model setting and did not explore jointly pretrained settings. Additionally, each participant watched a different movie while their sEEG activity was recorded, therefore the input stimuli varied widely across participants. In contrast, our study leverages video-audio models that capture the temporal events in videos, providing richer and more dynamic representations of the stimuli. By incorporating audio data, we preserve acoustic information that may be lost in text-based transcriptions. Moreover, we utilize fMRI data, which offers whole-brain coverage and higher spatial resolution, enabling a more comprehensive analysis of the brain activity. Our approach also considers both cross-modal and jointly pretrained multi-modal models, offering a more nuanced understanding of how different modalities interact and integrate information in the brain.

## K BASELINE ANALYSIS: SCRAMBLING INPUTS TO MULTI-MODAL MODELS

We conducted an additional baseline experiment where we kept the trained weights unchanged and shuffled the movie stimuli into a scrambled order as input to the two multi-modal models: cross-modal and jointly-pretrained models, as shown in Fig. 12 as "IB Concat Shuffle" and "TVLT Joint Shuffle" respectively.

From Fig. 12, we observe that embeddings from multi-modal models exhibit significantly better brain alignment compared to both randomly initialized models and when passing scrambled clips as input. Furthermore, when comparing scrambled input to pretrained models with randomly initialized models, the scrambled input shows improved alignment over random initialization. Overall, this sanity check confirms that representations from multi-modal models maintain meaningful alignment with brain activity, even when the stimulus order is scrambled, highlighting their robustness and effectiveness.

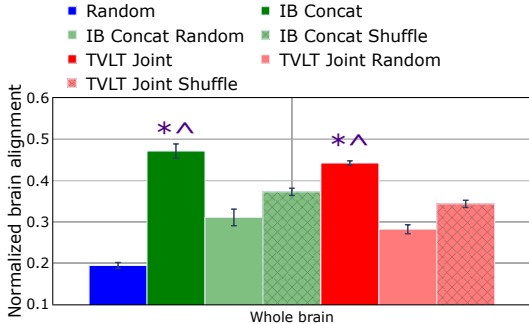

Figure 12: The plot compares the average normalized brain alignment across the whole brain under three conditions: (i) pretrained models, (ii) randomly initialized models, and (iii) pretrained multi-modal models using scrambled videos as input. Error bars represent the standard error of the mean across participants. ∗ indicates cases where multi-modal embeddings are significantly better than randomly initialized models, i.e., $p \leq 0.05$. ∧ indicates cases where multi-modal embeddings are significantly better than scrambled videos as input to multi-modal models, i.e., $p \leq 0.05$.

## L  Whole Brain, Language and Visual ROIs analysis: Shared and Unique variance between Multi-modal and Unimodal models

**Residual Analysis on the Brain:** The features can be removed from the model representations, or from the brain recordings. Conceptually, the results of these approaches should be the same because when the feature is removed completely from either the input or/and the target, it would not be able to further impact the observed alignment. However, practically, brain recordings are noisier than model representations and so estimating the removal regression model will be more difficult especially with fMRI data of low SNR. Thus residual analysis on the brain is less effective. Therefore, we opt to remove features from the model representations where we can exercise more control.

**Unique Variance Calculation:** We now build two voxelwise joint encoding models: (i) that includes both unimodal and cross-modal features, (ii) that includes both unimodal and jointly-pretrained features. Using these joint encoding models and prior encoding models, we compute the unique variance explained by each model i.e. unimodal and cross-modal, unimodal and jointly-pretrained models. Also, we compute the shared variance between these models. The results are shown in Figs. 13, 14 and 15 for the whole brain, language network and visual network, respectively. We make the following observations from Figs. 13, 14 and 15: (i) It can be clearly seen that the shared variance between jointly-pretrained (TVLT) model and the unimodal models is significantly lower than that observed between cross-modal (IB Concat) versus unimodal models. This result is in line with the earlier results on residual analysis where IB Concat showed larger drop in performance when unimodal information is removed as compared to that of TVLT model (see Fig. 4). Specifically, we can see that there is a larger drop in performance for removal of video features as compared to that with removal of speech features. (ii) For the visual network, as shown in Fig. 15, we observe that the unique explained variance between TVLT and unimodal VMs is comparable, while the cross-modal model IB Concat exhibits a higher unique explained variance compared to unimodal VMs. In contrast to unimodal VMs, both cross-modal and jointly pre-trained models demonstrate higher unique explained variance compared to unimodal SMs.

## M  Multi-modal versus unimodal effects

**Multi-modal effects:** In general, multi-modal models have better predictivity in the language regions (see Fig. 2).

**Unimodal effects:** Unimodal models have higher predictivity in the early sensory regions (visual and auditory). Such patterns of results are expected.

**Critical Differences:** However, critical differences are also observed. Although multi-modal models perform with around 50% alignment (Fig. 3) in high-level visual regions (PPA, MT), they seem to

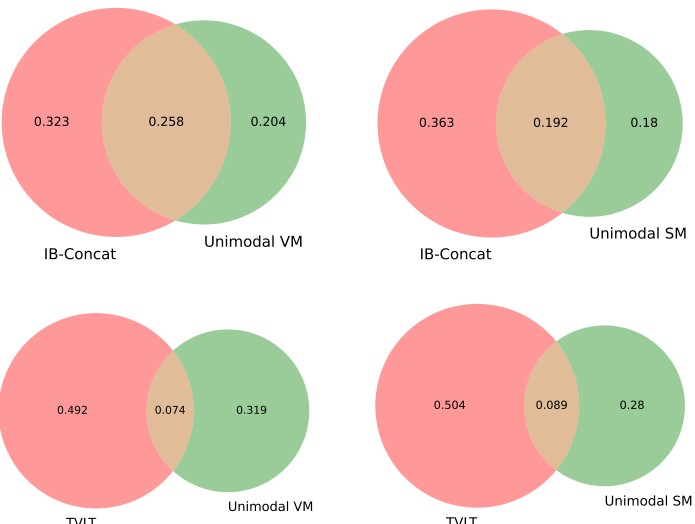

Figure 13: Whole Brain Analysis: Shared and Unique Variance explained between Cross-modal (IBConcat) and Unimodal models (VM, SM), Jointly-pretrained (TVLT) and Unimodal models (VM, SM). In each plot, Pink Area (Left Circle - Intersection) represents the unique variance explained by the multi-modal model that is not shared with the unimodal model. Green Area (Right Circle - Intersection) represents the unique variance explained by the unimodal model that is not shared with the Multi-modal model. Light Brown Intersection (Overlap) represents the shared variance between the multi-modal and unimodal model. It indicates the extent to which both models explain overlapping neural variance in the whole brain.

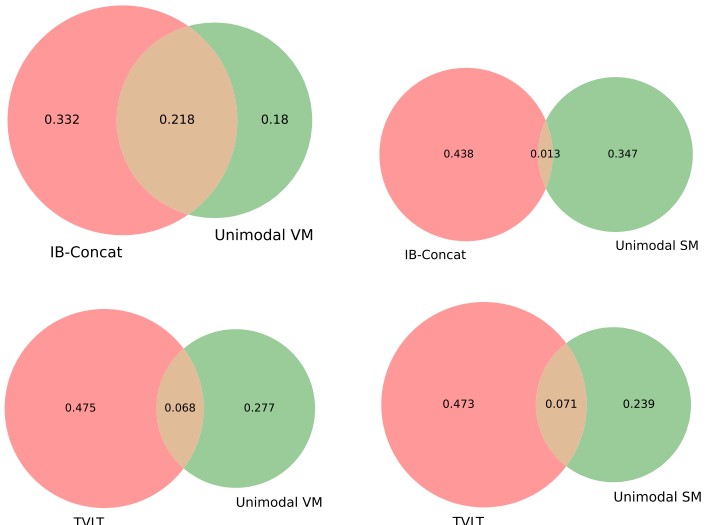

Figure 14: Language Network: Shared and Unique Variance explained between Cross-modal (IB Concat) and Unimodal models (VM, SM), Jointly-pretrained (TVLT) and Unimodal models (VM, SM). In each plot, Pink Area (Left Circle - Intersection) represents the unique variance explained by the multi-modal model that is not shared with the unimodal model. Green Area (Right Circle - Intersection) represents the unique variance explained by the unimodal model that is not shared with the Multi-modal model. Light Brown Intersection (Overlap) represents the shared variance between the multi-modal and unimodal model. It indicates the extent to which both models explain overlapping neural variance in the whole brain.

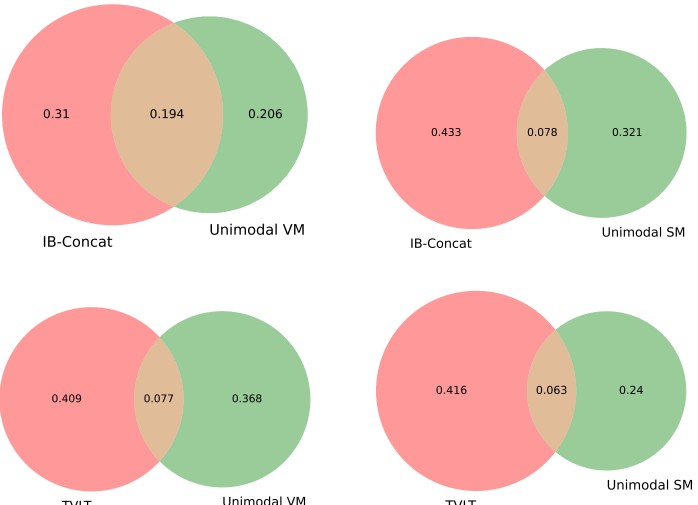

Figure 15: Visual Network: Shared and Unique Variance explained between Cross-modal (IB Concat) and Unimodal models (VM, SM), Jointly-pretrained (TVLT) and Unimodal models (VM, SM). In each plot, Pink Area (Left Circle - Intersection) represents the unique variance explained by the multi-modal model that is not shared with the unimodal model. Green Area (Right Circle - Intersection) represents the unique variance explained by the unimodal model that is not shared with the Multi-modal model. Light Brown Intersection (Overlap) represents the shared variance between the multi-modal and unimodal model. It indicates the extent to which both models explain overlapping neural variance in the whole brain.

perform less (around 40% alignment) in the early visual region (EVC). These patterns are also seen in the auditory regions (AC versus PTL). Thus, there seem to be both patterns of similarity as well as critical dissimilarities observed in the comparative analyses.

**Residual analysis:** For cross-modality models, the alignment in regions AG and MT is extremely high, and this alignment is only partially explained by video features (Fig. 4). This implies that significant unexplained alignment remains after the removal of video features. Conversely, the removal of speech features does not lead to a drop in brain alignment, indicating that there is additional information beyond speech features that is processed in these regions. The EVC region is well-predicted by unimodal video models, showing similar levels of predictivity with multi-modal models. A recent study by [Oota et al. 2024] explored the type of information in language models that predicts brain activity and found that low-level features in these models drive predictivity, such as speech-based language models predicting the visual cortex or text models predicting the auditory cortex.

When unimodal video model (VM) features are regressed out of multi-modal models, there is no performance drop in EVC but a drop in regions like PPA and LOC, suggesting that multi-modal models do not solely rely on corresponding unimodal features but also contain unexplained variance in EVC. Additionally, removing low-level features like motion energy may impact EVC performance. Interestingly, regressing out unimodal VM features does not affect speech-related information in multi-modal models, as speech models also exhibit brain predictivity in EVC.

# N   IMPACT OF DIVERSE MODEL ARCHITECTURES ON PERFORMANCE COMPARISON

Several prior brain encoding studies in the literature compared a variety of language/speech models (differing in their size, architecture, training dataset, etc.) and their brain alignment.

Schrimpf et al. (2021) investigated 43 language models ranging from distributed word embeddings models like Word2Vec, GloVe, FastText, to Sequence models such as RNN, LSTM, Contextualized models like ELMo, Transformer models like BERT, GPT-2 and Transformer-XL with its variations

such as base, small, and larger models. Although all these models have different architectures, training datasets, Schrimpf et al. (2021) considered each model as a subject and computed normalized brain predictivity, i.e., what percentage the model explains the variance given a ceiling value for each voxel. Similarly, Toneva & Wehbe (2019) used four different language models such as ELMo, BERT, USE and Transformer-XL and compared the explained variance of each model while doing brain alignment. Further, Aw & Toneva (2023) use four longer-context language models such as BART, Big-Bird, Longformer and Long-T5 and verify the deeper understanding of language models and brain alignment by the amount of variance explained by each model. Similarly, Antonello et al. (2021) used 101 language models including both pretrained and task-based language models and compared the amount of explained variance in the brain by extracting the semantic representations and whether these representations are closer to brain-level semantics. Recently, Oota & Toneva (2023); Oota et al. (2024a) used four text-based language models such as BERT, GPT-2, FLAN-T5 and BART, speech-based language models such as Wav2Vec2.0 and Whisper and verified the amount of explained variance in the brain at different language regions.

It is important to observe that all the above studies utilize a number of language models that are different in training architecture and training datasets, however the primary goal of all these studies is to investigate how close the semantic representations captured by each model aligns with brain-relevant semantics.

The extensive precedent in the literature, from studies comparing 43 models (Schrimpf et al., 2021) to those examining 101 models (Antonello et al., 2021), demonstrates that this approach is both valid and valuable for understanding the relationship between artificial and biological language processing.

## O    LIMITATIONS

The low alignment scores clearly show that despite the increasing popularity of multi-modal models in tackling complex tasks such as visual question answering, we are still far from developing a model that fully encapsulates the complete information processing steps involved in handling multi-modal naturalistic information in the brain. In the future, by fine-tuning these multi-modal models on specific tasks such as generating captions for videos, we can better leverage their alignment strengths. Further, multi-modal large language models (MLLMs) (Zhang et al., 2023; Ataallah et al., 2024; Wu et al., 2024) that align visual features from video frames into the LLM embedding space via a trainable linear projection layer, offer promise for enhanced multi-modal capabilities. Lastly, although we observe differences between the models (that we experimented with in this work) in terms of architectural variability and variability in pretraining methods, this suggests that future work could benefit from more tightly controlled comparisons to better isolate the effects of these factors. Addressing this limitation would provide deeper insights into how model design and training strategies impact brain alignment, particularly in the context of multi-modal stimuli datasets.

It is to be noted that there exist a relatively large number of vision-language or vision-alone or language-alone or speech-alone models as compared to video-audio models that is the primary focus of the current investigation. Therefore the current effort needs to take into consideration the sparsity of video-audio model availability. Despite this limitation, the current paper reports comparison across three video-only, two speech-only, one jointly-pretrained video-audio, and one cross-modal multi-modal model – to the best of our knowledge, this is by far the largest cohort of comparative analysis of multi-modal models and their alignment with brain representations resulting from multi-modal stimuli.

