# OpenReview forum: "Multi-modal brain encoding models for multi-modal stimuli"
_ICLR.cc/2025/Conference — ICLR 2025 Poster_

### Official Review · Reviewer_8Uj4 · 2024-10-24

**Soundness:** 3
**Presentation:** 3
**Contribution:** 3
**Rating:** 6
**Confidence:** 4

**Summary:**

The authors aim to address whether there is a difference in brain-alignment based on whether multimodal models had cross-modality pretraining (separate encoders for two modalities) or joint pretraining (shared encoder across modalities). They compare one model of each type to video and speech models and evaluate the prediction in a large-scale open access movie dataset. Using residual analyses, they investigate whether there are multi-modal representations in visual and language regions of the brain.

**Strengths:**

For the most part, the authors use standard and well validated methods to answer their question. I particularly like the approach of evaluating the performance on entire held-out videos and using the residual analysis to investing multi-modal effects above and beyond unimodal effects.

**Weaknesses:**

The authors only evaluate two multi-modal models and a small number of unimodal models. However, the chosen models differ on many factors (e.g. architecture, training data) in addition to the input modality and as a result, it is premature to draw conclusions about semantic representations in the brain that may be attributable to any of these factors. To my mind, there are two ways to mitigate this concern: 1) controlled model training and evaluation so that only one factor varies at a time, or 2) testing many different models of a given class such that even across significant model variations there is a robust effect of modality regardless of particular model factors. I think that this is a serious concern because not  all prior work has found that multi-modal models are more brain aligned. In controlled comparisons between visual models with the same architecture and training data, there was no performance increase as a result of contrastive image-language training (Conwell et al., 2023). These authors suggest that the higher alignment of CLIP relative to unimodal models in other work may be training set size.

Conwell, C., Prince, J. S., Kay, K. N., Alvarez, G. A., & Konkle, T. (2023). What can 1.8 billion regressions tell us about the pressures shaping high-level visual representation in brains and machines? (p. 2022.03.28.485868). bioRxiv. https://doi.org/10.1101/2022.03.28.485868

A minor weakness of the paper is that the authors use custom, non-standard acronyms and names for brain regions (e.g., scene visual area, SV, and object visual processing region, OV). It is confusing as a reader, but more critically, it difficult to understand what has been found for particular regions across the field, making the status of the literature more tenuous. I would suggest that the authors adopt standard acronyms throughout (e.g., PPA instead of SV).

Although the paper overall is fairly clear, section 6.3 and the corresponding figures (4, 9, and 10) are difficult to follow. I welcome clarification because, outside of a few lines in the discussion, I am having a hard time understanding which regions do show a multi-modal effect. Additionally, I think that the authors should emphasize whether they uncover expected unimodal effects in primary sensory cortices. In particular, we would expect that EVC would be predicted by visual models with no additional multi-modal contribution and similarly in AC but for auditory models. I am having trouble determining whether that is the case from the figures, but if it is not the case, it would lend more weight to my major concern about differences between models beyond modality.

**Questions:**

Why did the authors choose to use parametric statistics? To my knowledge non-parametric statistics are more common in NeuroAI to estimate a baseline performance rather than assuming one.

Are the brains on the bottom in Figure 5 the medial view? It is difficult to see why there are four brains in each box. Outside of labeling, showing the sulci on the inflated brains would help to orient the reader to what is being shown.

---

> ### Author Response · Authors · 2024-11-20
>
> *We thank the reviewer for their valuable comments and suggestions which are crucial for further strengthening our manuscript.*
>
> **Q1. Challenges in comparing relatively small number of models. Impact of diverse model architectures on performance comparison**
>
> Thank you for this interesting question.
> * We have provided a detailed discussion in our responses to **CQ1 and CQ2**. Kindly refer to those responses for more comprehensive information.
>
> **Q2. To my mind, there are two ways to mitigate this concern: 1) controlled model training and evaluation so that only one factor varies at a time, or 2) testing many different models of a given class such that even across significant model variations there is a robust effect of modality regardless of particular model factors.**
>
> Thank you for this valuable suggestion. We also acknowledge that testing a larger number of models within a given class can help determine if the effect of modality is robust across various model configurations.
>
> * In our study, we follow the second alternative focusing on models that belong to the same class, particularly those based on the Vision Transformer (ViT) architecture and the Audio Spectrogram Transformer (AST).
>
> **Unimodal Video Models:**
> * VideoMAE, ViViT, and ViT-H are all built upon the ViT architecture. These models share the same foundational structure, with variations primarily in training strategies or specific implementation details.
>   - VideoMAE: Employs masked autoencoders for video representation learning based on ViT.
>   - ViViT: Extends ViT to video by processing spatio-temporal tokens.
>   - ViT-H: A larger version of ViT with more parameters but the same architectural backbone.
>
> **Multimodal Models:**
> * ImageBind and TVLT are also based on the ViT architecture for the vision component and utilize AST for the audio component.
> * ImageBind: Binds multiple modalities by projecting them into a shared embedding space using ViT and AST encoders.
> * TVLT: Integrates visual and auditory information through ViT and AST, focusing on temporal alignment.
>
> * Thus, we believe that our current approach addresses your concern by utilizing multiple models within the same class, thereby reducing the impact of confounding factors related to architecture and training differences. By focusing on models based on ViT and AST, we provide a more controlled comparison that highlights the effect of input modality on brain alignment.
>
> **Q3. I think that this is a serious concern because not all prior work has found that multi-modal models are more brain aligned.**
>
> Thank you for pointing this out and for giving us the opportunity to clarify our findings in the context of existing research.
>
> **Multimodal Models and Brain Alignment:**
>
> * Recent brain encoding research has shown that multimodal models align more closely with brain activity than unimodal models, even when subjects are exposed to single-modality stimuli.
>
> * *[Tang et al. 2023]* investigated encoding models trained using representations from a multimodal transformer (BridgeTower) and a unimodal transformer (RoBERTa). They found that the multimodal transformer learned more aligned representations of concepts in both language and vision than unimodal ones.
> * *[Wang et al. 2023]* reported that CLIP, a multimodal model trained on image-text pairs, explains greater unique variance in higher-level visual areas compared to models trained only with image/label pairs (e.g., ResNet) or text-only models.
> * *[Oota et al. 2022]* examined four multimodal models—LXMERT, VisualBERT, CLIP, and ViLBERT—and found that these multimodal models showed improvements over the unimodal models in explaining brain activity.
> * *[Nakagi et al. 2024]* examined multimodal vision-semantic LLMs (GIT, BridgeTower and LLaVA-v1.5] that predict brain activity and find that these models uniquely capture representations in the association cortex better than unimodal models [BERT, DEiT, ResNet] combined.
> * Furthermore, **reviewer gD8j** suggested the following works: *Subramaniam et al. (2024)* and *Dong et al. (2023)*. These studies also demonstrate that multimodal models outperform both unimodal models and language-vision models with linearly integrated features, such as the concatenation of vision and language features.
>
> Our study extends this line of inquiry by examining brain encoding during multimodal stimuli (simultaneous audio and visual input), which more closely reflects real-world sensory experiences.
>
> *[Tang et al. 2023], Brain encoding models based on multimodal transformers can transfer across language and vision, NeurIPS-2023*
>
> *[Nakagi et al. 2024], Unveiling Multi-level and Multi-modal Semantic Representations in the Human Brain using Large Language Models, EMNLP-2024*
>
> *[Wang et al. 2023], Incorporating natural language into vision models improves prediction and understanding of higher visual cortex, Nature Machine Intelligence 2023*
>
> *[Oota et al. 2022], Visio-Linguistic Brain Encoding, COLING-2022*

---

> > ### Author Response · Authors · 2024-11-20
> >
> > **Q4. In controlled comparisons between visual models with the same architecture and training data, there was no performance increase as a result of contrastive image-language training (Conwell et al., 2023).**
> >
> > Thank you for pointing this out and for giving us the opportunity to clarify our findings in the context of existing research.
> >
> > **Addressing Conwell et al. (2023) Findings:**
> >
> > * Conwell et al. (2022) conducted controlled comparisons between visual models with identical architecture and training data, finding no performance improvement from contrastive image-language training. This outcome might be attributed to the evaluation metrics used in their analysis.
> > * Conwell et al. (2022) employed distributional similarity measures like Representational Similarity Analysis (RSA) even for voxel-wise encoding models. While these metrics are useful for comparing the overall statistical properties of neural and model representations, they may not capture detailed functional correspondences between specific model features and neural responses.
> > * In contrast, correlation-based metrics such as Pearson Correlation Coefficient (PCC) and explained variance are designed to assess the direct relationship between model predictions and neural activity on a voxel-by-voxel basis. These metrics are more sensitive to data nuances and can detect subtle alignments that distributional measures might miss *[Soni et al. 2024]*.
> > * For instance, *[Soni et al. 2024]* noted that using correlation-based metrics might emphasize linear relationships, while distance-based metrics could capture non-linear patterns. This sensitivity underscores the importance of carefully choosing the metric to ensure that the analysis aligns with the research objectives and accurately reflects the underlying data structures.
> > * The difference in evaluation metrics may partly explain why Conwell et al. (2022) did not observe a performance increase with contrastive image-language training, whereas other studies, including ours, have found that multimodal models show improved brain alignment by using correlation-based metrics that are sensitive to fine-grained functional correspondences.
> >
> > *[Soni et al. 2024] Conclusions about Neural Network to Brain Alignment are Profoundly Impacted by the Similarity Measure, Arxiv 2024*
> >
> >
> > **Q5. A minor weakness of the paper is that the authors use custom, non-standard acronyms and names for brain regions**
> >
> > Thank you for raising this important concern.
> > * We have updated the plots in our manuscript to include standard brain region names commonly used in the literature. Specifically, we have labeled the regions as follows:
> >   - PPA (Parahippocampal Place Area) for scene visual areas
> >   - OFA (Occipital Face Area) for face-related areas
> >   - LOC (Lateral Occipital Complex) for object-related regions
> > * These updates have been made both in the text and in the figures to enhance clarity and readability for readers. We believe that using these standard nomenclatures will make our findings more accessible and understandable to the scientific community.
> >
> > All these changes have been incorporated into the revised draft of the manuscript.

---

> > > ### Author Response · Authors · 2024-11-20
> > >
> > > **Q6. How do the authors clarify the multimodal effects in Section 6.3 and Figures 4, 9, and 10, particularly regarding whether expected unimodal effects are observed in primary sensory cortices**
> > >
> > > Thank you for this question. We understand the need for clarity as suggested by the reviewer. In the following we summarize the main results for the multi-modal and unimodal models.
> > >
> > > * Multi-modal effects:
> > >    - In general, multimodal models have better predictivity in the language regions (see Fig. 2).
> > > * Unimodal effects:
> > >    - Unimodal models have higher predictivity in the early sensory regions (visual and auditory). Such patterns of results are to be expected as the reviewer pointed out.
> > > * Critical Differences:
> > >    - However, critical differences are also observed. While multimodal models perform with around 50% alignment (Fig. 3) in high-level visual regions (PPA, MT), they seem to perform less (around 40% alignment) in the early visual region (EVC). These patterns are also seen in the auditory regions (AC versus PTL). Thus, there seem to be both patterns of similarity as well as critical dissimilarities observed in the comparative analyses.
> > > * Residual analysis:
> > >    - For cross-modality models, the alignment in regions AG and MT is extremely high, and this alignment is only partially explained by video features (Fig 4). This implies that significant unexplained alignment remains after the removal of video features. Conversely, the removal of speech features does not lead to a drop in brain alignment, indicating that there is additional information beyond speech features that is processed in these regions.
> > >
> > > * We agree with the reviewer that the EVC region is well-predicted by unimodal video models, showing similar levels of predictivity with multimodal models. A recent study by [Oota et al. 2024] explored the type of information in language models that predicts brain activity and found that low-level features in these models drive predictivity, such as speech-based language models predicting the visual cortex or text models predicting the auditory cortex.
> > > * When unimodal video model (VM) features are regressed out of multimodal models, there is no performance drop in EVC but a drop in regions like PPA and LOC, suggesting that multimodal models do not solely rely on corresponding unimodal features but also contain unexplained variance in EVC. Additionally, removing low-level features like motion energy may impact EVC performance. Interestingly, regressing out unimodal VM features does not affect speech-related information in multimodal models, as speech models also exhibit brain predictivity in EVC.
> > >
> > > **Q7. Why did the authors choose to use parametric statistics? To my knowledge non-parametric statistics are more common in NeuroAI to estimate a baseline performance rather than assuming one.**
> > >
> > > Thank you for this question. As explained in the Statistical Significance sub-section (lines 298-309 above Section 6), we describe the types of statistical testing employed. We employed a combination of both parametric and non-parametric methods to ensure robust and reliable results.
> > >
> > > * We use non-parametric tests such as permutation testing and Wilcoxson signed-rank test. Further, we also applied the Benjamini-Hochberg False Discovery Rate (FDR) correction for multiple comparisons. While FDR correction is often associated with parametric statistics, it is appropriate in our context because fMRI data is considered to have positive dependence, as established in previous research.
> > >
> > > **Q8. Are the brains on the bottom in Figure 5 the medial view? It is difficult to see why there are four brains in each box.**
> > >
> > > Thank you for this valuable feedback. We appreciate the opportunity to clarify the points regarding Figure 5: Hemispheres/Views: The top row represents lateral views, while the bottom row shows medial views of the brain.
> > >
> > > * We have now updated the figure caption in the revised draft to explicitly specify the views for clarity.
> > > * We have updated the brainmaps with one colormap (Reds) and colorbar to clearly indicate the percentage of decrease, with a range from 0% to 80%.
> > > * The colorbar shows the percentage decrease in brain alignment, where darker shade red voxels indicate a higher percentage decrease and white voxels indicate areas where unimodal video features do not contribute any shared information within the multi-modal context (i.e., no percentage drop).
> > > * We observe that the removal of unimodal video features from the cross-modal model (IB Concat) results in a significant performance drop of 40-50% in the visual regions. Additionally, a higher percentage drop is observed in the language regions (PTL and MFG) for the TVLT Joint model.
> > >
> > > Overall these edits ensure that the brainmaps now provide a more intuitive and clear visualization of the results.

---

> ### Comment · Reviewer_8Uj4 · 2024-11-22
>
> *Q2: Model variability*
>
> Overall, I find this explanation by the authors to somewhat mitigate my concern. However, I want to emphasize that belonging the same general class of architecture (e.g., ViT) is not the same as having the same architecture. The authors need to address this as a limitation in the main text of the paper.
>
> *Q4: Conwell et al. findings*
>
> I am not satisfied by the authors explanation of the Conwell findings. In controlled comparisons of models with and without language alignment, Wang et al. (2023) also did not find an increase in performance using an encoding approach as a result of language alignment throughout most of high-level visual cortex (Figure 5c). I will also emphasize here that my primary concern related to the Conwell and Wang findings is that when models are well controlled for architecture and training data, multimodality training may not have a meaningful effect. The authors can mitigate my concern here by just acknowledging that not all studies have found multimodal effects in the Related Work section and acknowledge that tightly controlled comparisons of models are needed in future work in the Limitations section (see *Q2*).
>
> *Q6: Multimodal effects*
>
> I appreciate the clarifications that the authors provided here. I would additionally like to see these clarifications reflected in the text.

---

> > ### Author Response · Authors · 2024-11-22
> >
> > Dear Reviewer 8Uj4,
> >
> > *We appreciate the reviewer’s positive feedback and are confident that it has contributed to enhancing the quality of our paper.*
> >
> > **Q2. Model Variability:**
> >
> > * Based on reviewer's suggestion, we have acknowledged this limitation in the revised draft. The Limitations section now emphasizes the importance of tightly controlled comparisons of models to better evaluate the effects of architecture and training data.
> >
> > **Q4: Conwell et al. findings**
> >
> > Thank you for this valuable suggestion.
> > * We have now included the works of Conwell et al. (2022) and Wang et al. (2022) in the Related Work section. These studies are discussed in the context of their findings that contrastive image-language training does not always lead to performance improvements, particularly in tightly controlled experiments.
> >
> > * Additionally, we have acknowledged the need for tightly controlled comparisons in future research in the Limitations section.
> >
> > **Q6. Multimodal effects.**
> >
> > Thank you for your positive feedback on clarification.
> > * We have now included Multimodal vs. Unimodal effects in the Appendix Section O.
> >
> > We hope these updates address your concerns and demonstrate our commitment to improving the manuscript. Thank you again for your insightful feedback.
> >
> > Please do let us know if there are any additional clarifications that we can offer. We would love to discuss more if any concern still remains. Otherwise, we would appreciate it if you could support the paper by increasing the score.
> >
> > regards,
> >
> > Authors

---

> > > ### Comment · Reviewer_8Uj4 · 2024-11-22
> > >
> > > Thank you for addressing these additional concerns! I have updated my score to reflect the incorporation of these changes.

---

> > > > ### Author Response · Authors · 2024-11-23
> > > >
> > > > We appreciate the reviewer's feedback and are confident that it has enhanced the paper's quality.

---

### Official Review · Reviewer_cWB1 · 2024-10-31

**Soundness:** 3
**Presentation:** 3
**Contribution:** 3
**Rating:** 6
**Confidence:** 3

**Summary:**

This paper compares different multimodal AI models to human brain responses while participants view audiovisual movies. They compare two different multimodal models, one cross-modal model that learns separate visual/audio embeddings and projects them into a shared representational space, and one jointly pretrained multimodal model, and three unimodal (vision or speech) models. The results show that in most brain regions multimodal training improves encoding model performance of voxel activity, particularly compared to unimodal speech models. The authors do additional residual analysis to understand the unique contribution of multimodal models (over unimodal models) to brain predictivity.

Overall, this is an interesting approach and the paper has many strengths. The link between results and overall conclusions was not always clear. In particular, the small number of highly varied models makes it difficult to draw strong conclusions about parallels between multimodal processing in the models and brains. Finally, there were several clarification/presentation questions.

**Strengths:**

There are many interesting and novel aspects to this paper. First, while there have been extensive encoding model studies on visual or audio stimuli, few have looked at model comparison to multimodal movies. The comparison of audiovisual models to this data is particularly novel. Further the comparison of different types of multimodal models is interesting (though it is difficult to draw strong conclusions about what their comparison tells you about multimodal processing in the brain, see below), and particularly the attempts to quantify what additional explanatory power a multimodal model has over unimodal models. The encoding analyses were also robust, particularly the cross-movie train/test splits.

**Weaknesses:**

The biggest issues stem from the comparison of the performance of a relatively small number of models that differ along many factors. This limitation makes it difficult to attribute any model differences to multimodality or different cross-modal training schemes, as the model architecture and training sets vary from model to model.

The fMRI dataset uses a small-n, data-rich design, which is good, but given this, it would help to see the results at the individual subjects’ level (in the appendix). On bar plots, it would be nice to plot each of the six subjects as a point overlaid on the average bar (rather than error bars which can obscure differences across the small number of subjects).

The residual analyses and results are somewhat confusing. The residual correlation with unimodal features was 0.56, which is still quite high. Given this, it is not clear that unimodal information was removed from multimodal models. Alternatively, the authors could do the residual analysis on the brain instead of the models (i.e., fit a model with both unimodal and cross modal and predict with just cross modal). Relatedly they could calculate the product measure or unique variance explained by multimodal models above unimodal models from this joint model.

Overall, the language and visual region responses look largely the same in most analyses. There are some quantitative/statistical differences, but the pattern is extremely similar. The authors should address this.

Perhaps related to the above point, all regions of interest are defined anatomically, but there is a fair amount of subject-wise variability in the anatomy of high-level functional regions, such as language. The authors should address this limitation.

Acronyms and plots were difficult to follow. Would help to spell out throughout vision versus lang regions for example, and clarify the legend in figure 4 (e.g., it was not clear on first read that “-“ indicates “minus”).

Figure 5 is difficult to read and interpret. In terms of clarification, the authors should specify hemispheres/views (it looks like a lateral view on top, medial on bottom, but I’m not certain). The results look extremely noisy and seem to show random patterns, with as much red as blue. Blue are areas the unimodal models perform better? How should that be interpreted? The legend says the colorbar indicates “percentage increase/decrease. Does this mean 0.5% or 50%? If the former, these are very tiny differences, which perhaps explains the above confusion, but I believe makes it difficult to draw any strong conclusions about these results.

I had questions about two of the conclusions listed in the discussion. I was unsure what the second part of conclusion 2 (“This variance can be partially attributed to the removal of video features alone, rather than auditory features.”) meant. I am also unconvinced of conclusion #4 given the overall similarity between language and visual brain regions described above.

Typos:
-	Line 224: “audo” --> “audio”
-	Line 276: “wmploy” --> “employ”

**Questions:**

It would help to have additional methodological details in some sections:
-	Was cross-subject prediction generated by using all of the predictor subjects voxels, to predict the target subject’s voxel-wise responses?
-	What are the six different modalities in the image-bind modality? I thought it was an audio-visual model (which I would consider two modalities)
-	The layer-wise predictions for models are shown in the appendix, but do the main text figures use all layers or best layer? If the latter, how is this layer selected.
-	Are the whole brain results averaged across all cortical voxels? Or only those with above-threshold cross-subject prediction? Or some other criteria?

---

> ### Author Response · Authors · 2024-11-20
>
> *We thank the reviewer for their valuable comments and suggestions which are crucial for further strengthening our manuscript.*
>
> **Q1. Challenges in comparing relatively small number of models. Impact of diverse model architectures on performance comparison**
>
> Thank you for this interesting question.
> * We have provided a detailed discussion in our responses to **CQ1 and CQ2**. Kindly refer to those responses for more comprehensive information.
>
> **Q2. The fMRI dataset uses a small-n, data-rich design, which is good, but given this, it would help to see the results at the individual subjects’ level (in the appendix). On bar plots, it would be nice to plot each of the six subjects as a point overlaid on the average bar**
>
> We thank the reviewer for this suggestion.
> * For brain encoding studies using naturalistic fMRI datasets, we follow the common practices in this area of research in that the number of samples per participant is more important than the number of subjects (six) because the predictive models are trained independently for each participant. Thus, having more samples per participant helps us learn a better predictive model. The Movie10 fMRI dataset is one of the largest datasets in terms of samples per participant (~12,950 samples), and that is the main reason for its frequent use. Our results also clearly show that this dataset is sufficient to learn a good predictive model, as we can predict up to 50% of the explainable variance for held-out brain recordings that were not used for training (e.g., Fig 2). Hence, our fMRI dataset is not small.
>
> * Based on the reviewer’s suggestion, **we have updated the bar plots in Appendix Figs. 7, 8,  9 and 10** to display each subject’s predictivity score as individual points instead of using error bars. This visualization provides a clearer representation of variability across subjects, allowing readers to assess individual differences more effectively. We believe these changes address the reviewer's concerns and enhance the clarity and interpretability of our results.
>
> **Q3. The residual analyses and results are somewhat confusing.  Alternatively, the authors could do the residual analysis on the brain instead of the models. Relatedly they could calculate the product measure or unique variance explained by multimodal models above unimodal models from this joint model.**
>
> Thank you for this interesting question.
>
> **Residual Analysis on the Brain:**
> * The features can be removed from the model representations (as we do), or from the brain recordings (as suggested by the reviewer).
> * Conceptually, the results of these approaches should be the same because when the feature is removed completely from either the input or/and the target, it would not be able to further impact the observed alignment.
> * However, practically, brain recordings are noisier than model representations and so estimating the removal regression model will be more difficult especially with fMRI data of low SNR.
> * Thus residual analysis on the brain is less effective.  Therefore, we opt to remove features from the model representations where we can exercise more control, similar to *[Toneva et al, 2022; Oota et al., 2023; Oota et al., 2024]*. We will clarify this reasoning in the main paper.
>
> **Unique Variance Calculation:**
> * Based on reviewer’s suggestion, we now build two voxelwise joint encoding models:
>   - (i) that includes both unimodal and cross-modal features,
>   - (ii)  that includes both unimodal and jointly-pretrained features.
>   - Using these joint encoding models and prior encoding models, we compute the unique variance explained by each model i.e. unimodal and cross-modal, unimodal and jointly-pretrained models.
>   - Also, we compute the shared variance between these models.
>   - The results, presented in **Appendix N, Figures 12, 13 and 14**, for the whole brain and the language network, reveal that the shared variance between the jointly pretrained (TVLT) model and the unimodal models is significantly lower than that observed between the cross-modal (IBConcat) model and the unimodal models.
>   - This finding aligns with earlier residual analysis results, where IBConcat showed a larger performance drop when unimodal information was removed compared to the TVLT model (see Figure 4). Specifically, we observe a more pronounced performance drop when video features are removed compared to when speech features are removed.
>   - For the visual network, as shown in Figure 14,  we observe that the unique explained variance between TVLT and unimodal VMs is comparable, while the cross-modal model IB Concat exhibits a higher unique explained variance compared to unimodal VMs. In contrast to unimodal VMs, both cross-modal and jointly pre-trained models demonstrate higher unique explained variance compared to unimodal SMs.
>
> *[Toneva et al. 2022], Combining computational controls with natural text reveals aspects of meaning composition, Nature Computational Science 2022*

---

> > ### Author Response · Authors · 2024-11-20
> >
> > **Q4. Overall, the language and visual region responses look largely the same in most analyses. There are some quantitative/statistical differences, but the pattern is extremely similar. The authors should address this.**
> >
> > * It is true that the overall pattern of results with respect to how multimodal versus unimodal models behave across the language and visual regions are similar.
> > * In general, multimodal models seem to have better predictivity in the language regions (see Fig. 2).
> > * Similarly, unimodal models have higher predictivity in the early sensory regions (visual and auditory). Such patterns of results are to be expected.
> > * However, critical differences are also observed. While multimodal models perform with around 50% alignment, they seem to perform less (around 40% alignment) in the early visual region (EVC) but seem to improve in higher visual areas (SV (rerenamed as PPA), MT).
> > * These patterns are also seen in the auditory regions (AC versus PTL). Thus, there seem to be both patterns of similarity as well as critical dissimilarities observed in the comparative analyses.
> >
> > **Q5. All regions of interest are defined anatomically, but there is a fair amount of subject-wise variability in the anatomy of high-level functional regions, such as language.**
> >
> > Thank you for this interesting question. The reviewer is concerned with subject-wise variability in the anatomy of high-level functional regions. To address this limitation, we follow several important steps:
> >
> > **Dataset Size:** The dataset we use, Movie10, is one of the largest in terms of samples per participant (~12,950 samples), which is the main reason for its frequent use. In comparison, naturalistic fMRI datasets used in linguistic brain encoding studies typically have around 9,000 samples (e.g. Moth-Radio-Hour) with six subjects, which is still fewer than the Movie10 dataset.
> >
> > **High-Spatial Resolution:** We use fMRI datasets that provide high-spatial resolution and more detailed, individualized maps of functional regions, unlike EEG datasets where source estimation is more challenging.
> >
> > **Cross-Subject Predictivity:** Our analyses focus on cross-subject predictivity, which provides shared explained variance among participants. The number of samples per participant is crucial because our predictive models are trained independently for each participant. More samples per participant help us learn better predictive models.
> >
> > **Aggregated Results:** We present results at the whole brain, language, and visual ROIs levels by aggregating normalized brain predictivity results. This approach helps us account for individual differences more effectively.
> >
> > * Overall, the normalized brain alignment computed per participant helps in assessing how closely our model predictions explain variance through brain predictivity. Our results clearly show that this dataset is sufficient to learn a good predictive model, as we can predict up to 50% of the explainable variance for held-out brain recordings when averaged across participants.
> > * Despite all these, we do agree with the reviewer that the inherent uncertainties in individual functional localization need to be acknowledged as a limitation in all such brain alignment studies.
> >
> > **Q6. Acronyms and plots were difficult to follow. Would help to spell out throughout vision versus lang regions**
> >
> > Thank you for this valuable suggestion.
> > * We have updated the plots to clearly specify the Regions of Interest (ROIs) as follows: Language: ROI, Visual: ROI, and Auditory: ROI. We hope this distinction helps clarify the figures.
> > * Additionally, we have revised Figures 4, 9, and 10 to explicitly indicate that the "–" symbol represents residuals. These updates aim to improve the readability and interpretability of the plots.

---

> > > ### Author Response · Authors · 2024-11-20
> > >
> > > **Q7. Figure 5 is difficult to read and interpret. In terms of clarification, the authors should specify hemispheres/views. The results look extremely noisy  How should that be interpreted?**
> > >
> > > Thank you for this valuable feedback. We appreciate the opportunity to clarify the points regarding Figure 5:
> > > **Hemispheres/Views:** The top row represents lateral views, while the bottom row shows medial views of the brain.
> > > * We have now updated the figure caption in the revised draft to explicitly specify the views for clarity.
> > > * We have updated the brainmaps with one colormap (Reds) and colorbar to clearly indicate the percentage of decrease, with a range from 0% to 80%.
> > > * The colorbar shows the percentage decrease in brain alignment, where darker shade red voxels indicate a higher percentage decrease and white voxels indicate areas where unimodal video features do not contribute any shared information within the multi-modal context (i.e., no percentage drop).
> > > * We observe that the removal of unimodal video features from the cross-modal model (IB Concat) results in a significant performance drop of 40-50% in the visual regions. Additionally, a higher percentage drop is observed in the language regions (PTL and MFG) for the TVLT Joint model.
> > >
> > > * Overall these edits ensure that the brainmaps now provide a more intuitive and clear visualization of the results.
> > >
> > > **Q8. I had questions about two of the conclusions listed in the discussion.**
> > >
> > > * Conclusion 2 is based on our observation made in lines 460-468, reproduced here:
> > > “For cross-modality models, the alignment in regions AG and MT is extremely high, and this alignment is only partially explained by video features. This implies that significant unexplained alignment remains after the removal of video features.
> > > * Conversely, the removal of speech features does not lead to a drop in brain alignment, indicating that there is additional information beyond speech features that is processed in these regions.”
> > >
> > > Please see the response to Q5 above for patterns of dissimilarity observed for the language and visual regions.
> > >
> > > **Q9. Typos: - Line 224: “audo” --> “audio” - Line 276: “wmploy” --> “employ”**
> > >
> > > Thank you for pointing this out. We have corrected the identified typos in the revised draft.
> > >
> > > **Q10. Was cross-subject prediction generated by using all of the predictor subjects voxels, to predict the target subject’s voxel-wise responses? **
> > >
> > > Thank you for this question.
> > > * Yes, similar to previous studies *[Schrimpf et al. 2021] [Oota et al. 2024] [Alkhamiss et al. 2024]*, cross-subject predictions were generated by using all possible combinations of s participants (s∈[2,6]), where voxel-wise responses from s−1 predictor participants were used to predict the target participant's voxel-wise responses.
> > >
> > > *[Schrimpf et al. 2021]  The neural architecture of language: Integrative modeling converges on predictive processing. PNAS, 2021*
> > >
> > > *[Oota et al. 2024] Speech language models lack important brain relevant semantics, ACL 2024*
> > >
> > > *[Alkhamissi et al. 2024] Brain-Like Language Processing via a Shallow Untrained Multihead Attention Network, Arxiv 2024*
> > >
> > > **Q11. What are the six different modalities in the image-bind modality? I thought it was an audio-visual model (which I would consider two modalities) -**
> > >
> > > Thank you for this question.
> > > The six modalities in the ImageBind model are:
> > >   - Image/Video: Visual modalities, including static images and videos.
> > >   - Text: Natural language data.
> > >   - Audio: Sound and acoustic signals.
> > >   - Inertial Measurement Units (IMU): Motion and activity data captured via sensors.
> > >   - Depth: Spatial depth information from sources like LiDAR or stereo cameras.
> > >   - Thermal: Heat or temperature patterns from infrared sensors.
> > >
> > > * During the training of the ImageBind model: Image and text encoders are kept frozen. Audio, depth, thermal, and IMU encoders are updated.
> > > * This architecture results in a model with a total of 132 million parameters, enabling cross-modal embeddings across these six diverse modalities.
> > >
> > > **Q12. The layer-wise predictions for models are shown in the appendix, but do the main text figures use all layers or best layer? If the latter, how is this layer selected.**
> > >
> > > Thank you for this question.
> > > * The figures in the main text present the average normalized brain alignment across subjects, layers, and voxels. This approach ensures that the results are representative of overall model performance, without bias toward any specific layer. Note that we are only averaging across voxels which have a statistically significant brain alignment.

---

> > > > ### Author Response · Authors · 2024-11-20
> > > >
> > > > **Q13. Are the whole brain results averaged across all cortical voxels? Or only those with above-threshold cross-subject prediction? Or some other criteria?**
> > > >
> > > > Thank you for this question.
> > > > * We considered the whole-brain voxels based on the following process for obtaining normalized brain alignment:
> > > > * Voxel Selection: We initially selected voxels with a cross-subject predictivity> 0.05 Pearson correlation, in line with previous works (*[Popham et al., 2021], [La Tour et al., 2022], [La Tour et al., 2023]*).
> > > > * Normalization: Each voxel prediction was divided by its corresponding cross-subject predictivity.
> > > > * Averaging: The normalized predictions were then averaged across all selected voxels to compute the whole-brain results.

---

> > > > > ### Comment · Reviewer_cWB1 · 2024-11-21
> > > > >
> > > > > Thank you for these responses and updates. I think the revisions help to clarify the paper, particularly the figures, and I appreciate the additional variance partitioning analysis the authors conducted.
> > > > >
> > > > > My primary concern, however, still holds: the comparison of the performance of a relatively small number of models that differ along many factors. This limitation makes it difficult to attribute any model differences to multimodality or different cross-modal training schemes, as the model architecture and training sets vary from model to model.
> > > > >
> > > > > I appreciate the approach of treating individual models as data points, and that any one such model factor may not make a difference across a large set of models. However, the difference between any pair of models in a set can be large and due to a variety of factors we don’t fully understand. I think this is an important limitation to acknowledge.
> > > > >
> > > > > That said, I think this is an interesting paper, and in particular the use of and analysis of different components of multimodal models is an important advance in the field. I have adjusted my score accordingly.
> > > > >
> > > > > Finally, I want to clarify my comment on the fMRI dataset: I agree the dataset is not small, and I did not say this in my review. In my review I referred to it as a small-N dataset (meaning few subjects, many samples). I think this is preferable to a large-N, low-sample dataset, however, it often lends itself better to individual analysis and plots. Thank you for including these.

---

> > > > > > ### Author Response · Authors · 2024-11-21
> > > > > >
> > > > > > Dear Reviewer cWB1,
> > > > > >
> > > > > > We appreciate the reviewer’s positive feedback and are confident that it has contributed to enhancing the quality of our paper.
> > > > > >
> > > > > > We acknowledge the point raised regarding the differences between the pairs of models in terms of variability in architectures. As suggested by the reviewer. we will include this point in the limitations section. Additionally, we suggest that future work could focus on more controlled comparisons to better isolate the effects of these factors, as model-controlled experiments are longstanding questions for the linguistic community.
> > > > > >
> > > > > > Regards,
> > > > > > The Authors

---

> > > > > > > ### Author Response · Authors · 2024-11-22
> > > > > > >
> > > > > > > Dear Reviewer cWB1,
> > > > > > >
> > > > > > > We appreciate the reviewer’s positive feedback and are confident that it has contributed to enhancing the quality of our paper.
> > > > > > >
> > > > > > > Based on the reviewer's suggestion, we have addressed this limitation in the revised draft. The Limitations section now highlights the importance of conducting tightly controlled model comparisons to more effectively evaluate the impact of architecture and training data.
> > > > > > >
> > > > > > > Regards,
> > > > > > >
> > > > > > > Authors

---

### Official Review · Reviewer_gD8j · 2024-11-04

**Soundness:** 3
**Presentation:** 3
**Contribution:** 3
**Rating:** 8
**Confidence:** 4

**Summary:**

- Existing work uses unimodal models to identify language and vision processing pathways in the brain. This paper studies multimodal processing in the brain. Multimodal networks are aligned to the brain, and regions with better alignment are identified as the sites of multimodal processing. To verify that the alignment is actually due to multimodality, unimodal influence is removed from multimodal representations by subtracting out the multimodal target, as predicted by the unimodal input, from the multimodal features.

**Strengths:**

- This work is novel in that it is the first to use fMRI. But other works also take similar approaches to identifying multimodal processing (see weaknesses).
- Findings of the difference between cross-modal and jointly-trained models with respect to regions is novel
- These findings have neuroscientific significance
- Appropriate random baselines are used to give context to alignment numbers is given

**Weaknesses:**

- Relationship with previous work [1] and similar concurrent work [2,3] that uses multimodal models to probe for multimodal processing in the brain should be discussed. [1] studies cross-modal and jointly trained networks and uses a naturalistic multi-modal stimuli.
- The random baseline described in 6.1 is good to make sure that the trained model weights matter. To better get a sense of whether the alignment actually reflects processing in the brain, another good sanity check would be to run a permutation test in which you keep the trained weights, but give the movie stimulus inputs to the model in scrambled order. This would give a floor for the scale of alignments that we see in subsequent results.

## Small things
- Line 276 typo: wmploy -> employ

## References
[1] Subramaniam, V., et al. "Revealing Vision-Language Integration in the Brain with Multimodal Networks." International Conference on Machine Learning. International Conference on Machine Learning (ICML), 2024.

[2] Kewenig, Viktor, et al. "Evidence of Human-Like Visual-Linguistic Integration in Multimodal Large Language Models During Predictive Language Processing." arXiv preprint arXiv:2308.06035 (2023).

[3] Dong, Dota Tianai, and Mariya Toneva. "Vision-Language Integration in Multimodal Video Transformers (Partially) Aligns with the Brain." arXiv preprint arXiv:2311.07766 (2023).

**Questions:**

- For the procedure described in the Figure 1b caption on removing unimodal influence: why subtract out the unimodal contribution? Why not learn a regression directly from the unimodal contribution itself, i.e., the predictions of $r$?
- Can it be said that the IB-audio and IB-video unimodal representations described on lines 223-224 are not truly unimodal, since they are extracted from a model that was trained with multimodal inputs? Then, they reflect correspondences between language and vision.
- Figure 3 second row, middle two columns: Why does the green bar not have $\wedge *$ for SV? It seems significantly higher than both light green bars. Why does the green bar have $\wedge *$ for MT? It only seems significantly higher than on light green bar.

---

> ### Author Response · Authors · 2024-11-20
>
> *We thank the reviewer for their strong positive, insightful and valuable comments and suggestions which are crucial for further strengthening our manuscript.*
>
> **Q1. Relationship with previous works that uses multimodal models to probe for multimodal processing in the brain should be discussed.**
>
> Thank you for this question. We would like to clarify that our study is different from previous studies in the following aspects:
>
> * **Subramaniam et al. (2024)** utilized vision-language models based on image frame-text pairs, despite the stimuli being continuous movies. While effective for certain tasks, this approach may overlook the temporal dynamics inherent in videos. Additionally, their use of stereoencephalography (SEEG), although offers high temporal resolution, is limited in spatial resolution and typically restricted to specific brain regions where the electrodes are implanted. Finally, their study focused on multimodal integration in a cross-modal model setting and did not explore jointly pretrained settings. Additionally, each participant watched a different movie while their SEEG activity was recorded, therefore the input stimuli varied widely across participants.
>   - In contrast, our study leverages video-audio models that capture the temporal events in videos, providing richer and more dynamic representations of the stimuli. By incorporating audio data, we preserve acoustic information that may be lost in text-based transcriptions. Moreover, we utilize fMRI data, which offers whole-brain coverage and higher spatial resolution, enabling a more comprehensive analysis of the brain activity.
>   - Our approach also considers both cross-modal and jointly pretrained multimodal models, offering a more nuanced understanding of how different modalities interact and integrate information in the brain.
> * **Dong et al. (2023)** utilized the Friends web series fMRI dataset to investigate the effectiveness of pretrained versus fine-tuned multimodal video transformers using video+text stimuli-based brain activity. Their study specifically examined the impact of fine-tuning a multimodal model on brain alignment, comparing performance before and after fine-tuning. However, they did not explore cross-modal vs. jointly pretrained model analysis or the comparison between multimodal and unimodal models, leaving it unclear which type of multimodal models perform best for predicting brain activity. In contrast, our study focuses on video+audio stimuli and includes a comprehensive residual analysis, providing deeper insights into the contributions of different modalities.
> * **Kewenig et al. (2024)** conducted a study involving 200 human participants who provided ratings while watching short audio-visual clips (6 seconds each) to estimate the predictability of upcoming words. This study specifically focused on behavioral evidence, demonstrating that multimodal attention models can leverage contextual information to predict upcoming words in a manner more aligned with human behavior. Notably, this study did not involve brain recordings; instead, the authors collected behavioral ratings and focused on human attention as indexed through eye-tracking data.
>   - These differences suggest that our study is better placed to provide deeper insights into how the brain processes and integrates multimodal information, leveraging the strengths of video-audio models leveraging the comprehensive spatial coverage of fMRI.
>
> We have included this extended related work in **Appendix J** of the revised paper.
>
> **Q2. Baseline Analysis: Scrambling Inputs to Multimodal Models**
>
> Thank you for suggesting this new experiment.
>
> * Based on the reviewer’s suggestion, we conducted an additional baseline experiment where we kept the trained weights unchanged and shuffled the movie stimuli into a scrambled order as input to the two multimodal models: cross-modal and jointly-pretrained models. The results of these experiments have been added to **Appendix K, Fig. 12** of the revised paper.
> * The results demonstrate that embeddings from multimodal models exhibit significantly better brain alignment compared to both randomly initialized models and when passing scrambled clips as input. Furthermore, when comparing scrambled input to pretrained models with randomly initialized models, the scrambled input shows improved alignment over random initialization.
> * Overall, this sanity check confirms that representations from multimodal models maintain meaningful alignment with brain activity, even when the stimulus order is scrambled, highlighting their robustness and effectiveness.
>
> **Q3. Line 276 typo: wmploy -> employ**
>
> Thank you for pointing this out. We have corrected the identified typos in the revised draft.

---

> > ### Author Response · Authors · 2024-11-20
> >
> > **Q4. For the procedure described in the Figure 1b caption on removing unimodal influence: why subtract out the unimodal contribution? Why not learn a regression directly from the unimodal contribution itself, i.e., the predictions of r?**
> >
> > Thank you for this question.
> > * We would like to clarify that in this analysis, we did not directly subtract the unimodal contribution from multimodal models.
> >   - Instead, we first learn a linear mapping from unimodal to multimodal features using ridge regression (r).
> >   - This function (r) estimates the unimodal aspect correlating with the multimodal representations (CM(X)).
> >   - Now, we compute the residuals, |CM(X) - r(VM(X))|, where CM(X) represents the multimodal contribution and VM(X) represents the unimodal contribution.
> >   - The residuals allow us to characterize any changes in brain alignment performance when unimodal representations are removed. The subtraction operation here conceptually stands for removal of unimodal contributions from multimodal representations and may not be interpreted as direct subtractions.
> > * Further, if we perform regression of r(VM(X)) with the brain responses, as the reviewer suggests, this would be similar (identical) to looking at unimodal brain alignments.
> > * These results are already presented in Fig. 3 (Orange and Blue bars for Video and Speech unimodal models, respectively).
> >
> > **Q5. Can it be said that the IB-audio and IB-video unimodal representations described on lines 223-224 are not truly unimodal, since they are extracted from a model that was trained with multimodal inputs? Then, they reflect correspondences between language and vision.**
> >
> > Thank you for this question.
> > * Yes, it is true that IB-audio and IB-video are also multimodal representations. These embeddings are derived from the pretrained ImageBind model, which inherently generates multimodal representations. By "modality-specific embeddings," we refer to the extraction of audio and video embeddings from ImageBind. While these embeddings are modality-specific in their origin (audio or video), they are still multimodal in nature due to the model's multimodal pretraining, as the reviewer points out.
> > * Just to add, Unimodal VM (ViT-B, VideoMAE, ViViT) and Unimodal SM (Wav2Vec2.0, AST) are the pure unimodal models.
> >
> > **Q6. Figure 3 second row, middle two columns: Why does the green bar not have ^\* for SV? It seems significantly higher than both light green bars. Why does the green bar have ^\* for MT? It only seems significantly higher than on light green bar.**
> >
> > Thank you for raising this question.
> > * To clarify, the * symbol indicates cases where multi-modal embeddings are significantly better than unimodal video models. For the SV region (renamed as PPA, Parahippocampal Place Area), IB Concat (green bar) is not significantly better than unimodal video models (orange bar).
> > * ∧ indicates cases where multi-modal embeddings are significantly better than unimodal speech models. For PPA region, IB Concat (green bar) is significantly better than unimodal speech models. As a result, we use only the ^ symbol to denote this relationship.
> > * We have already included this information in the caption for clarity. A similar explanation applies to the MT region, where the multi-modal embeddings show a similar pattern of significance.

---

> > > ### Author Response · Authors · 2024-11-23
> > >
> > > Dear Reviewer gD8j,
> > >
> > > We appreciate your feedback and effort you have invested in evaluating our work.
> > >
> > > In response to your insightful comments, we have addressed the issues you highlighted. We believe these revisions significantly contribute to the clarity and completeness of the paper. We kindly request you to verify our response and consider updating your evaluation based on the revisions made.
> > >
> > > Should you have any further questions or suggestions, we are ready to provide additional information or clarification as needed.
> > >
> > > Thanks for your help

---

> ### Comment · Reviewer_gD8j · 2024-11-25
>
> I thank the authors for their thorough response. I am more convinced that the work covers novel ground, based on their explanations of prior work. I have upgraded my score accordingly.

---

> > ### Author Response · Authors · 2024-11-25
> >
> > We appreciate the reviewer's positive feedback and are confident that it has enhanced the paper's quality.

---

### Author Response · Authors · 2024-11-20
**Common Responses to Reviewers cWB1 and 8Uj4:**

*We are grateful to all reviewers for their strong positive feedback, time and their constructive suggestions, which will further strengthen the impact of our work.*

**CQ1: Challenges in comparing relatively small number of models (Reviewers: cWB1, 8Uj4)**

Thank you for this interesting question. We would like to provide a clear discussion as follows.

* We appreciate the concern raised by the reviewer about performance comparison across a small cohort of models.
* It is to be noted that there exist a relatively large number of vision-language or vision-alone or language-alone or speech-alone models as compared to video-audio models that is the primary focus of the current investigation. Therefore the current effort needs to take into consideration the sparsity of video-audio model availability.
* Despite this limitation, the current paper reports comparison across three video-only, two speech-only, one jointly-pretrained video-audio, and one cross-modal multi-modal model – to the best of our knowledge, this is by far the largest cohort of comparative analysis of multi-modal models and their alignment with brain representations resulting from multi-modal stimuli.
* Now, let us discuss the state-of-the-art practices in brain alignment experiments in CQ2.

We have added this discussion in **Appendix L** of the revised paper.

**CQ2. Impact of diverse model architectures on performance comparison (Reviewers: cWB1, 8Uj4)**

Several prior brain encoding studies in the literature compared a variety of language/speech models (differing in their size, architecture, training dataset, etc.) and their brain alignment.

* *[Schrimpf et al. 2021]* investigated 43 language models ranging from distributed word embeddings models like Word2Vec, GloVe, FastText, to Sequence models such as RNN, LSTM, Contextualized models like ELMo, Transformer models like BERT, GPT-2 and Transformer-XL with its variations such as base, small, and larger models. Although all these models have different architectures, training datasets, *[Schrimpf et al. 2021]* considered each model as a subject and computed normalized brain predictivity, i.e., what percentage the model explains the variance given a ceiling value for each voxel.
* Similarly, *[Toneva et al. 2019]* used four different language models such as ELMo, BERT, USE and Transformer-XL and compared the explained variance of each model while doing brain alignment.
* Further, *[Aw et al. 2023]* use four longer-context language models such as BART, Big-Bird, Longformer and Long-T5 and verify the deeper understanding of language models and brain alignment by the amount of variance explained by each model.
* Similarly, *[Antonello et al. 2022]* used 101 language models including both pretrained and task-based language models and compared the amount of explained variance in the brain by extracting the semantic representations and whether these representations are closer to brain-level semantics.
* Recently, *[Oota et al. 2023] [Oota et al. 2024]* used four text-based language models such as BERT, GPT-2, FLAN-T5 and BART, speech-based language models such as Wav2Vec2.0 and Whisper and verified the amount of explained variance in the brain at different language regions.

It is important to observe that all the above studies utilize a number of language models that are different in training architecture and training datasets, however the primary goal of all these studies is to investigate how close the semantic representations captured by each model aligns with brain-relevant semantics.

The extensive precedent in the literature, from studies comparing 43 models *[Schrimpf et al. 2021]* to those examining 101 models *[Antonello et al. 2022]*, demonstrates that this approach is both valid and valuable for understanding the relationship between artificial and biological language processing.

We have added this discussion in **Appendix M** of the revised paper.

*[Schrimpf et al. 2021], The neural architecture of language: Integrative modeling converges on predictive processing. PNAS 2021*

*[Toneva et al. 2019], Interpreting and improving natural-language processing (in machines) with natural language-processing (in the brain). NeurIPS 2019*

*[Antonello et al. 2022], Low-dimensional structure in the space of language representations is reflected in brain responses, NeurIPS 2022*

*[Aw et al. 2023], Training language models to summarize narratives improves brain alignment, ICLR 2023*

*[Oota et al. 2024], Speech language models lack important brain relevant semantics, ACL 2024*

---

### Author Response · Authors · 2024-11-20
**Summary of our responses and revision**

**We are grateful to all reviewers for their strong positive feedback, time and their constructive suggestions, which will further strengthen the impact of our work.**

**Summary of Reviewer Strengths:**

1. Novelty of Approach: This work is the first to use fMRI to investigate differences in cross-modal and jointly-trained multimodal models with respect to brain regions, a novel contribution with neuroscientific significance. **(reviewers: gD8J, cWB1)**
2. Significance of Findings: This study provides novel insights into the differences between cross-modal and jointly-trained models and their explanatory power for multimodal processing in the brain. **(reviewers: gD8J, cWB1)**
3. Well designed experiments: **(reviewers: gD8J, cWB1, 8Uj4)**
   - Authors use standard and well-validated methods were employed effectively, providing confidence in the study's findings.
   - The use of random baselines contextualizes alignment numbers, improving interpretability.
   - Cross-movie train/test splits and residual analyses add robustness to the encoding analyses, offering a more comprehensive evaluation of multimodal effects beyond unimodal contributions.
4. Innovative Comparisons: **(reviewers: gD8J, cWB1)**
	- The comparison between different types of multimodal models (e.g., audiovisual and unimodal) is interesting

**Additional changes to the draft during the rebuttal process**

We have updated the main manuscript and the appendix to address these following comments. The changes made in the manuscript are highlighted in blue color. The major additional changes are listed below.

1. **Extended Related Works** (Reviewer gD8j, 8Uj4):  We discuss how our current study is different from previous studies [Subramaniam et al. (2024)], [Kewenig et al. (2024) ], [Dong et al. (2023)] in the following aspects:
   - Subramaniam et al. (2024) employed vision-language models based on image-text pairs, potentially overlooking the temporal dynamics of continuous movies, and used SEEG, which is limited in spatial resolution and coverage. Their study focused on cross-modal integration without exploring jointly pretrained models, with each participant viewing different movie stimuli, leading to varied inputs across participants.
   - Kewenig et al. (2024) focused on behavioral evidence, demonstrating that multimodal attention models can leverage contextual information to predict upcoming words in a manner aligned with human behavior. Notably, the study did not involve brain recordings; instead, the authors collected behavioral ratings and focused on human attention as indexed through eye-tracking data.
   - Dong et al. (2023) study compared brain alignment performance before and after fine-tuning but did not explore jointly-pretrained mutlimodal models vs. multiple unimodal models, leaving open questions about which multimodal approaches best predict brain activity.
   - Conwell et al. (2022) and Wang et al. (2022): These studies are discussed in the context of their findings that contrastive image-language training does not always lead to performance improvements, particularly in tightly controlled experiments.
We have added results of these experiments in **Appendix J** of the revised paper.

2. **Baseline Analysis: Scrambling Inputs to Multimodal Models** (Reviewer gD8j):
   - we conducted an additional baseline experiment where we kept the trained weights unchanged and shuffled the movie stimuli into a scrambled order as input to the two multimodal models: cross-modal and jointly-pretrained models.
   - The results demonstrate that embeddings from multimodal models exhibit significantly better brain alignment compared to both randomly initialized models and when passing scrambled clips as input.

3. **Whole Brain, Language and Visual ROIs analysis: Shared and Unique variance between Multimodal and Unimodal models** (Reviewer cWB1): To empirically verify residual analysis, we now build two voxelwise joint encoding models: (1) that includes both unimodal and cross-modal features, (2)  that includes both unimodal and jointly-pretrained features.
   - Using these joint encoding models and prior encoding models, we compute the unique variance explained by each model i.e. unimodal and cross-modal, unimodal and jointly-pretrained models.
   - The results, presented in Appendix N, Figures 12, 13 and 14, for the whole brain and the language network, reveal that the shared variance between the jointly pretrained (TVLT) model and the unimodal models is significantly lower than that observed between the cross-modal (IBConcat) model and the unimodal models.

4. **Updation of Brainmap Plots and Figures in Appendix** (Reviewer cWB1): The revised appendix includes updates to Brainmap plots and figures (Figs. 7, 8, 9 and 10), featuring standard ROI names. Additionally, subject values are now represented as individual points on bar plots instead of using error bars, ensuring clearer visualization of subject-specific data.

---

### Meta-Review · Area_Chair_dY76 · 2024-12-20

**Metareview:**

The paper considers the problem of finding alignments between active regions of the brain from fMRI signals when a person watches multimodal content (audiovisual movies) and multimodal features extracted from state-of-the-art deep learning models. Correlation results using the Movie-10 dataset presents many interesting conclusions, especially regarding better alignment of multimodal models against unimodal ones, including better alignment of video models than speech/audio.

The paper received favorable reviews, with one accept and two borderline accepts. The reviewers appreciated the novelty of finding correlations between fMRI data and multimodal/unimodal features and the extensive experiments.

**Additional Comments On Reviewer Discussion:**

There were three major points on which the reviewers raised concerns.
1. On the small number of multi-modal models used in the study, each of these models differing in their capabilities over diverse factors, thus questioning the reliability/significance of the correlations derived (cWB1, 8Uj4),
2. Inconsistencies / lack of clarity in the some of the provided results (gD8j, cWB1)
3. Inconsistencies with the conclusions derived in prior methods that also explored multi-modal models for brain alignment (8Uj4).

For 1., authors argued that in the setting that is explored in this paper, the provided study is by far the largest cohort of models being explored, albeit relatively smaller. Authors have acknowledged this point in their limitations.

For 2., authors provided additional details clarifying the issues pointed out, including providing new results when scrambling video inputs as the reviewer suggested, and clarifying the differences to prior works.

For 3., authors clarify that they are extending the study in previous works examining brain region activations for multi-modal stimuli. They also agree to consider tightly-controlled experiments that are needed to derive similarities to previous studies.

Overall, the paper is well-written and makes a valuable contribution to the field. While there are questions that are difficult to be addressed within the scope of the current work, AC thinks the conclusions derived in the paper are sufficiently novel and thus recommends acceptance.

---

### Decision · Program_Chairs · 2025-01-22

Accept (Poster)